



# Estimating Ice Water Content and Snowfall Rate from radar measurements in the G-band

Karina McCusker[1], Chris Westbrook[1], Alessandro Battaglia[2,3], Kamil Mroz[4], Benjamin M. Courtier[2], Peter G. Huggard[5], Hui Wang[5], Richard Reeves[5], Christopher J. Walden[5,6], Richard Cotton[7], Stuart Fox[7], and Anthony J. Baran[7,8]

[1]Department of Meteorology, University of Reading, Reading, UK
[2]University of Leicester, Leicester, UK
[3]Politecnico of Turin, Turin, Italy
[4]European Centre for Medium-Range Weather Forecasts (ECMWF), Reading, UK
[5]RAL Space, STFC Rutherford Appleton Laboratory, Didcot, OX11 0QX, UK
[6]National Centre for Atmospheric Science, Leeds, UK
[7]Met Office, FitzRoy Road, Exeter, EX1 3PB, UK
[8]School of Physics, Astronomy and Mathematics, University of Hertfordshire, Hatfield, AL10 9AB, UK

**Correspondence:** Karina McCusker (k.mccusker@reading.ac.uk)

**Abstract.** We present theory and simulations to show that at frequencies of order 200 GHz (G-band) the radar cross section ($\sigma_r$) of ice particles larger than $\sim$ a quarter wavelength ($0.375 \, \mathrm{mm}$) is nearly directly proportional to their mass ($m$), and hence measurements of radar reflectivity ($Z$) at this frequency are directly proportional to the ice water content (IWC), with no other assumptions about the shape or breadth of the particle size distribution required. For the same reason, vertically pointing

Doppler velocities at this frequency provide the mass-weighted mean vertical velocity of the particles, and the product of $Z$ with the mean Doppler velocity (MDV) is proportional to the snowfall rate ($S$). This presents the opportunity for straightforward but accurate retrievals of ice microphysics.

We explore the sensitivity of such retrievals to the scattering model for ice particles. We find that all seven models examined, four with random orientation and three with horizontal orientation, have $\sigma_r \propto m$ in this regime, but that the coefficient of

proportionality varies between models. The dominant factor controlling this coefficient is the mass-size relationship for the ice particles, and specifically the mass of a wavelength-sized ice particle. If this information is known, or can be assumed, then the ice population parameters above can be retrieved with high accuracy. For mass-weighted mean diameters $D_m > 0.5 \, \mathrm{mm}$ the variation in the IWC$-Z$ relationship is within $\approx 30\%$, and the variation in the $S-(Z \times \mathrm{MDV})$ relationship is within $\approx 15\%$.

The method is applied to retrieve IWC and $S$ during two case studies, with measurements from the GRaCE 200 GHz Doppler

radar at Chilbolton Observatory in the UK. In the first of these case studies, retrieved snowfall rates from particles falling aloft in a precipitating ice cloud were compared to gauge data at the surface. In the second case study, retrieved ice water contents from a deep non-precipitating stratiform ice cloud were compared to measurements made using an evaporative water content probe on board the Facility for Airborne Atmospheric Measurements (FAAM) BAe-146 instrumented research aircraft. In both cases a statistical comparison was necessary because of imperfect colocation of the radar measurements and in-situ/gauge





sampling. The retrievals fall within the distributions from the retrieved water content and snowfall fields, and follow consistent trends with time (Case 1) and height (Case 2), providing evidence that this method produces realistic retrievals.

Application of the same technique at even higher radar frequencies would allow clouds with smaller particles (e.g. in high altitude cirrus clouds) to be characterised. Because of the increased gaseous attenuation at such frequencies, the latter may be more practical from airborne or spaceborne platforms.

## 1   Introduction

Recent technological advancements have enabled the development of G-band (110-300 GHz) radars with high enough sensitivities for atmospheric remote sensing. There have been a number of successful ground-based and airborne demonstrators such as VIPR in USA (Cooper et al., 2018, 2021; Millán et al., 2024; Roy et al., 2020, 2022; Lamer et al., 2021), GRaCE in UK (Courtier et al., 2022, 2024, in prep.), CloudCube in USA (Socuellamos et al., 2024a, b, c), and more recently GRaWAC in Germany (Bühler et al., 2025). As hypothesised by Battaglia et al. (2014), the demonstrator instruments have shown that measurements in the G-band contain valuable information for a variety of atmospheric scenarios. In terms of humidity and liquid clouds, G-band data has been used successfully to profile water vapour (Roy et al., 2020) and, when combined with other frequencies, to retrieve small amounts of liquid water in warm shallow clouds (Socuellamos et al., 2024c) and improve microphysical retrievals in light rain (Courtier et al., 2024). In the ice phase, Lamer et al. (2021) and McCusker et al. (in prep.) highlight that using a dual-frequency pair inclusive of G-band data allows sizing of smaller ice particles than has been possible using lower frequencies such as Ka and W bands.

Although there is currently no G-band cloud radar in space, with the highest frequency space-borne radars operating in the W-band (on-board CloudSat (Marchand et al., 2008) and the Doppler radar on EarthCARE (Illingworth et al., 2015)), the aforementioned findings suggest that a spaceborne G-band radar could be on the horizon. Indeed, GRaCE and CloudCube are intended as demonstrators for future satellite missions. Ideally such missions will exploit the benefits of multi-frequency observations; however it is also important to explore simpler methods, which may be accessible using a single frequency, or provide a robust first estimate of the microphysical state of the cloud before refinement using dual-frequency ratios (which typically have higher uncertainty). The success of EarthCARE's Doppler capability (Kim et al., 2025) also motivates us to consider the information content of Doppler measurements, in addition to reflectivity.

Algorithms that currently exist to retrieve ice water content (IWC) and snowfall rate ($S$) from radar measurements can be of varying complexity, but the simplest methods are direct statistical relationships between reflectivity and ice population properties, i.e. $Z-$IWC or $Z-S$ relationships. These relationships may be used directly to retrieve particle properties, or may form the initial estimate for an optimal estimation retrieval (this is the case, for example, in the EarthCARE cloud and



precipitation microphysics (C-CLD) algorithm (Mroz et al., 2023), where the estimate is subsequently refined using the Doppler
    velocity information).

    However, it is well documented that the relationship between $Z$ and microphysical parameters of interest is not unique, and
    varies considerably (e.g. Protat et al. (2016), Matrosov and Heymsfield (2008)). Brown et al. (1995) show IWC as a function
    of $Z$ at W-band, computed from a large number of in-situ particle size distributions. The IWC for a given reflectivity is spread

over approximately an order of magnitude across the various cloud samples, with typical uncertainties on the order of a factor
    of two, and this variation is driven by variations in the breadth and shape of the particle size distribution (PSD), i.e., how wide
    or narrow the range of particle sizes is, and how the particle sizes are distributed around the mean. Retrieving $S$ from reflectivity
    presents similar problems, with order of magnitude variability in the value of $S$ for a given $Z$ (Hiley et al. 2011, Fuller et al.
    2023 and references therein). Because $Z$ and IWC (or $Z$ and $S$) are different moments of the PSD, their interrelationship

is sensitive to what that PSD shape and width is. Interestingly, the variability in the $\mathrm{IWC} - Z$ relationship decreases as $Z$
    increases, due to increased non-Rayleigh scattering (Brown et al., 1995) and it is this phenomenon that we will exploit in the
    current study.

    In this manuscript, we explore the usefulness of radar reflectivity and Doppler velocity in the G-band for estimating IWC
    and $S$, with a view towards a future spaceborne G-band radar. We consider theory and simulations, along with measurements

from a ground-based G-band radar, to determine what information is required for accurate retrievals. We show that IWC and
    $S$ can be retrieved with a single frequency G-band radar provided that the mass of a wavelength-sized particle is known or
    can be assumed, while the details of the PSD breadth and shape are not required. Two case studies are provided to illustrate
    the practical application of the theory, and to demonstrate that the retrievals are consistent with in-situ measurements of IWC
    and $S$. This work presents the first known retrievals of ice cloud and snowfall properties using a G-band radar, representing a

major step forward in the use of high-frequency radar for atmospheric remote sensing. Unlike traditional optimal estimation
    approaches, which are computationally intensive but widely adopted, the method introduced here is both computationally
    efficient and robust, offering a practical alternative without compromising reliability.

## 2   Parameters of interest

Our first measurement parameter is the equivalent radar reflectivity factor $Z$:


$$Z = 10^{18} \frac{\lambda^4}{\pi^5 |K_{\mathrm{water}}|^2} \int\limits_0^\infty N(D)\sigma_r(D)\mathrm{d}D, \tag{1}$$

and we denote the logarithmic equivalent as dBZ$= 10\log_{10}(Z)$. $N(D)$ is the particle size distribution (PSD, m$^{-4}$), such that
there are $N(D)\mathrm{d}D$ ice particles with maximum dimension in the interval $[D, D + \mathrm{d}D]$, and $\sigma_r(D)$ is the radar cross section
(m$^2$) of each ice particle of that size. $\lambda$ is the free space wavelength, 1.5 mm at 200 GHz. The factor $10^{18}$ is present to convert

the SI units of $N$, $\sigma_r$ and $\lambda$ to the conventional reflectivity unit of mm$^6$ m$^{-3}$. We choose to normalise $Z$ using $|K_{\mathrm{water}}|^2 = 0.93$,
    which is the value for liquid water at cm-wavelengths, following the approach of Hogan et al. (2006).



In addition to $Z$, we can also consider the mean Doppler velocity (MDV), which is the average of the vertical velocity of the particles $v(D)$ (equal to the still-air fall speeds of the particles, plus any vertical air motion), weighted by their radar cross sections and PSDs, i.e.

$$\text{MDV} = \frac{\int_0^\infty N(D)\sigma_r(D)v(D)\mathrm{d}D}{\int_0^\infty N(D)\sigma_r(D)\mathrm{d}D}. \tag{2}$$

Our retrieval parameters are IWC and $S$, which can be written in terms of the PSD as:

$$\text{IWC} = \int\limits_0^\infty N(D)m(D)dD \tag{3}$$

$$S = \int\limits_0^\infty N(D)m(D)v(D)dD, \tag{4}$$

where $m$ is the particle mass (in kg).

Note that the equations above give IWC with units of $\mathrm{kg\,m^{-3}}$ and $S$ with units of $\mathrm{kg\,m^{-3}m\,s^{-1}}$. Thus to express IWC in $\mathrm{g\,m^{-3}}$ and $S$ in $\mathrm{mm\,h^{-1}}$ (as we do in the following section) requires multiplication by $10^3$ and 3600 respectively.

As we will see shortly, at high frequencies such as G-band, $Z$ becomes almost directly proportional to IWC, while the product $(Z \times \text{MDV})$ becomes nearly directly proportional to $S$. This is in stark contrast to the behaviour at lower frequencies, such as Ka-band, and is a result of the non-Rayleigh scattering which occurs when the particle becomes comparable in scale to the wavelength.

## 3 Simulations of ice and snow parameters using realistic particle scattering models

In this section, we perform simulations of IWC and $S$ using Eqs. 3 and 4. To illustrate the concept, we use the Large Plate Aggregate mixture from the ARTS scattering database (Eriksson et al., 2018), which combines pristine plate habits to cover the smaller sizes of the PSD with aggregates of plates to represent the larger sizes. The particles are randomly oriented in 3D, and details on their mass can be found in Sect. 5 and Table 1. We assume an exponential PSD: $N(D) = N_0 \exp(-\Lambda D)$ where the slope parameter $\Lambda$ controls the breadth of the distribution (and hence the average particle size), while the intercept parameter $N_0$ scales the particle concentrations up and down to allow IWC to vary independently of particle size.

### 3.1 Ice Water content

Simulations of IWC are presented in Fig. 1. The left column shows IWC vs dBZ at Ka-, W-, and G-band.

The three different linestyles show results using $N_0 = 10^6$, $10^7$, $10^8$ $\mathrm{m^{-4}}$. It is clear from panel (a) that at lower frequencies (Ka-band), the relationship between IWC and $Z$ varies considerably with $N_0$, while the sensitivity to PSD parameters decreases systematically with frequency (panels (c) and (e)). For example, consider a dBZ measurement of $0$ dBZ. The IWC corresponding to this value is highly sensitive to $N_0$ in the Ka-band, with a difference of 137% between $N_0 = 10^6$ and $N_0 = 10^8$ $\mathrm{m^{-4}}$. The





difference in IWC between $N_0 = 10^6$ and $N_0 = 10^8$ m$^{-4}$ decreases with frequency. There is a smaller (but still considerable) difference of 52% in the W-band, while the difference in the G-band is much smaller, at 7%.

The right column shows the ratio IWC/$Z$ for different values of the mass-weighted mean particle diameter

$$D_m = \frac{\int_0^\infty N(D)m(D)DdD}{\int_0^\infty N(D)m(D)dD}. \tag{5}$$

At Ka-band, the ratio IWC/$Z$ decreases continuously with $D_m$ (panel b). This means there would be large uncertainties
associated with retrieval algorithms based on a direct relation between $Z$ and IWC, because $D_m$ is unknown a-priori. At higher frequencies, the variation in IWC/$Z$ becomes smaller, particularly at larger $D_m$. Panel (d) shows that in the W-band IWC/$Z$ steadily decreases at small $D_m$, reaching a minimum at around 2mm before slightly increasing again at large $D_m$. At G-band (panel f), the ratio approaches an almost constant value for sufficiently large $D_m$.

The variation in IWC/$Z$ with $D_m$ in the interval 0.5–2 mm is a factor of 13 at Ka-band, a factor of 3 at W-band, but only a
factor of 1.4 at G-band (corresponding to a 33% variation relative to the mean). In other words, a direct retrieval of IWC from $Z$ in the G-band would have considerably lower uncertainties than at lower frequencies.

In this illustration we have used a single scattering model. The sensitivity to this choice is explored further in Sect. 5.

## 3.2  Snowfall Rate

Fig. 2 shows simulations related to the snowfall rate S. The left column shows the product $Z \times MDV$ for a range of $S$ values.
As above, the different linestyles show results using different values of $N_0$. The results mirror those for IWC shown in Fig. 1, with the sensitivity of the simulations to $N_0$ decreasing with frequency. For a measurement of $Z \times MDV = 1$ mm$^6$m$^{-3}$ms$^{-1}$, the difference in $S$ between $N_0 = 10^6$ and $N_0 = 10^8$ m$^{-4}$ is 122%, 24%, and 13% for Ka-, W-, and G-band, respectively.

The right column shows the ratio $S/(Z \times \mathrm{MDV})$ for a range of $D_m$. These simulations also mirror those presented for IWC/$Z$, whereby the ratio decreases with $D_m$ in the Ka-band but becomes more constant at large $D_m$ in the G-band. The
$S/(Z \times \mathrm{MDV})$ ratio varies by a factor of 1.2 for $D_m$ between 0.5 and 2 mm in the G-band (corresponding to a 15% variation relative to the mean), while for Ka- and W- bands the ratio varies by factors of 11 and 2.3 respectively, which again suggests that $S$ can be retrieved from measurements of $Z$ and MDV in the G-band.

## 4  Theory

In the previous section, G-band scattering simulations showed that the ratios IWC/$Z$ and $S/(Z \times \mathrm{MDV})$ approach a near
constant value for values of $D_m$ greater than $\approx 0.5$ mm. In this section, we will consider the theory behind those results.

In the Rayleigh regime (i.e. where $D \ll \lambda$), the radar cross section scales in proportion to the square of the volume of ice within the particle (Doviak and Zrnić, 1993):

$$\sigma_r = 36\pi^3 \lambda^{-4} \frac{m^2}{\rho_{ice}^2} \left| \frac{\epsilon - 1}{\epsilon + 2} \right|^2 c_{ns}, \tag{6}$$




**Figure 1.** Simulations using the Large Plate Aggregate mixture from the ARTS database, assuming exponential PSDs. The different rows from top to bottom show simulations in the Ka-, W-, and G-band. The left column shows dBZ for a range of IWC, where the different linestyles represent different values of $N_0$ in the PSDs. The right column shows the ratio IWC/Z for a range of $D_m$.

where $m$ is the particle mass, $\rho_{ice}$ is the density of solid ice, and $\epsilon$ is the complex dielectric constant of solid ice. As a result, in
that regime, we have:

$$Z_{\text{Rayleigh}} = C_{\text{Rayleigh}} \int_0^\infty N(D) c_{ns} m(D)^2 \mathrm{d}D, \qquad (7)$$





**Figure 2.** As in Fig. 1, but here the left column shows $Z \times \mathrm{MDV}$ for a range of $S$, and the right column shows the ratio $S/(Z \times \mathrm{MDV})$ for a range of $D_m$.

where

$$C_{\mathrm{Rayleigh}} = 10^{18} \frac{36 |K_{ice}|^2}{0.93 \pi^2 \rho_{ice}^2} \tag{8}$$



and $|K_{ice}|^2$ is the dielectric factor for ice, which is $\approx 0.174$ at cm and mm wavelengths (Hogan et al., 2006; Westbrook
et al., 2007; Hogan et al., 2017). The dimensionless coefficient $c_{ns}$ is a function of the shape of the particle, and its orientation
relative to the polarisation of the radar, and is of order unity. For spherical particles $c_{ns} = 1$. The Rayleigh Gans Approximation
(RGA, see McCusker et al. 2019 and references therein) also assumes that $c_{ns} \approx 1$, reflecting its assumption that the scatterer
is composed of a weak dielectric, and hence that the electric field incident on any part of an ice particle is equal to the
applied field from the radar. In reality, the coupling between neighbouring parts of the ice particle enhances the mean electric
field (McCusker et al., 2019), and increases $c_{ns}$ to values larger than one. For example, Hogan et al. (2017) suggest that for
aggregates of non-spherical ice crystals, $1 < c_{ns} < 2$. In what follows, we will assume that $c_{ns}$ is independent of $D$ and can be
taken outside the integral in Eq. 7.

Evidently $Z_{\text{Rayleigh}}$ does not scale in proportion to IWC. To convert from one to the other, it is therefore necessary to
independently estimate the shape and width of the PSD (and to know the relationship between $m$ and $D$).

There is extensive literature (e.g. Locatelli and Hobbs (1974); Mitchell et al. (1990)) that suggests ice particle mass and
maximum dimension are connected to one another statistically, via a power law relationship, i.e. $m = aD^b$. Such power law
relationships for aggregates and complex crystals are indicative of a geometry that is statistically fractal, with fractal dimension
equal to $b$. For aggregates, the exponent $b$ is typically $\approx 2$, and this leads to $Z_{\text{Rayleigh}} \propto \int_0^\infty D^4 N(D) \mathrm{d}D$, while IWC $\propto$
$\int_0^\infty D^2 N(D) \mathrm{d}D$. Again, this emphasises that at low frequencies (where scattering is close to the Rayleigh regime), $Z$ is a
significantly higher moment of the PSD than the ice water content or snowfall rate.

Equation 7 is appropriate for scattering by ice particles at centimetre wavelengths; however at millimetre wavelengths non-
Rayleigh effects become increasingly important. In this case, the radar cross section has a more complex dependence on particle
size. Physically, as the dimensions of the particle become comparable to the wavelength, interference begins to occur between
the waves that are scattered from different parts of the particle. We express this non-Rayleigh behaviour via a dimensionless
factor $f$ which depends on the size of the particle relative to the wavelength:

$$Z = C_{\text{Rayleigh}} c_{ns} \int_0^\infty N(D) m(D)^2 f(D/\lambda) \mathrm{d}D. \tag{9}$$

For fractal aggregates, Sorensen (2001) used RGA to deduce the scaling of $f$ when the particle is large enough compared to
the wavelength, and found that it had a power law dependence, which for backscattering is:

$$f \approx c_f \left( \frac{4\pi R_g}{\lambda} \right)^{-b}. \tag{10}$$

Here $R_g$ is the radius of gyration, which as we will show later is proportional to $D$, and the non-dimensional coefficient $c_f$ is
of order unity. Berg (2008) used the discrete dipole approximation (DDA) to confirm that this power law scaling is still evident
even when RGA is not strictly applicable (such as is the case for ice: e.g. McCusker et al. (2021)). Inserting this into Eq. 9, we
obtain:

$$Z \approx C_{\text{Rayleigh}} c_{ns} c_f (4\pi c_{R_g})^{-b} \lambda^b \int_0^\infty N(D) m(D)^2 D^{-b} \mathrm{d}D, \tag{11}$$





where $c_{R_g} = (R_g/D)$, which as we will demonstrate later has a typical value of around 0.3. Equation 11 is valid provided that the particles dominating the reflectivity are sufficiently large relative to the wavelength. Based on our simulations in figures 1 and 2, this occurs when $D_m \gtrsim 0.5$ mm. Inserting the mass-size relationship $m = aD^b$ into Eqs. 11 and 3, we obtain our key result, valid in the regime where the particles dominating the radar measurements are comparable to, or larger than the wavelength:

$$Z \approx \kappa m_\lambda \int_0^\infty N(D) a D^b \mathrm{d}D = \kappa m_\lambda \mathrm{IWC}, \tag{12}$$

where the coefficient $\kappa = C_{\mathrm{Rayleigh}} c_{ns} c_f (4\pi c_{R_g})^{-b}$ and $m_\lambda = a\lambda^b$ is the mass of a wavelength-sized particle i.e. $m(D = \lambda)$.

   In the same way, we can show that

$$(Z \times \mathrm{MDV}) \approx C_{\mathrm{Rayleigh}} c_{ns} c_f (4\pi c_{R_g})^{-b} \lambda^b \int_0^\infty N(D) m(D)^2 D^{-b} v(D) \mathrm{d}D = \kappa m_\lambda S. \tag{13}$$

It also follows that the mean Doppler velocity is equal to the mass-weighted vertical velocity of the snowflakes.

The results in Eqs. 12 and 13 explain the near-constant values of $\mathrm{IWC}/Z$ and $S/(Z \times \mathrm{MDV})$ for large $D_m$ at G-band in Figs. 1(f) and 2(f). The key parameters in these relationships are therefore $m_\lambda$ and $\kappa$, and we now investigate these factors in depth.

## 5   Estimation of $m_\lambda$, $\kappa$, and its components

From Eqs. 12 and 13, we expect $\mathrm{IWC}/Z$ and $S/(Z \times \mathrm{MDV})$ to approach a constant value $\mathcal{A} = 1/(\kappa m_\lambda)$ if $D_m$ is large
enough (note that the numerical values of the two ratios differ due to unit conversions, as discussed below). This is backed up by the simulations which showed that curves of $\mathrm{IWC}/Z$ and $S/(Z \times \mathrm{MDV})$ are increasingly independent of $D_m$ if particle size is large enough compared to the wavelength, i.e. $D_m \gtrsim 0.5$ mm in the G-band. We have seen that $\kappa$ is composed of the coefficients $c_{ns}$, $c_f$, $c_{R_g}$ which we may expect to vary between scattering models as well as the exponent $b$ of the mass-size relationship. The other key parameter is $m_\lambda$ which depends purely on the mass-size parameters $a$ and $b$ along with the (known)
radar wavelength. In what follows we estimate the various components of $\kappa$, and tabulate $a, b, m_\lambda$ for a number of scattering models, to determine which parameter the results are most sensitive to, i.e. what properties we need to know to perform accurate retrievals.

   Table 1 provides estimates for the mass-size parameters $a$ and $b$, as well as the coefficients $c_{ns}$, $c_{Rg}$ and $c_f$ for a number of different shape/scattering models. As well as the Large Plate Aggregate mixture, we consider three other mixtures from
the ARTS database, namely the Large Block Aggregate mixture, Large Column Aggregate mixture, and the ICON snow mixture. These are described in more detail in Ekelund et al. (2020). These different mixtures cover a range of different mass-size relationships, with the block aggregate mixture producing the heaviest particles for a given $D$, and the column aggregate mixture producing the lightest particles. Note that since we are mainly interested in particles larger than a few hundred micrometres in size, the values of $a$ and $b$ provided in Table 1 correspond to the aggregates in the mixture (see Table



5.3 in Ekelund et al. (2021)), rather than the whole mixture of monomers and aggregates. The particles in all four ARTS mixtures are randomly oriented in 3D.

In addition to the ARTS mixtures, we also analysed data for three habits from the database of Mroz and Leinonen (2023), which includes both unrimed and rimed aggregates of dendrites. Here we include results for the unrimed aggregates, along with aggregates with an effective liquid water path (ELWP) of 0.1 and 0.2 $\mathrm{kgm}^{-2}$. The ELWP corresponds to what the liquid water

path would be, if riming were 100% efficient. In other words, larger ELWP corresponds to particles which are more heavily rimed. In contrast to the ARTS database, the Mroz and Leinonen (2023) data includes a large number of particle realisations scattered across a broad range of sizes. In our analysis, we took the particle properties and scattering data from this large ensemble, and rebinned them into uniform size bins. The values of $a$ and $b$ provided in the table for these models were then estimated by fitting a power law function to the average mass $m$ in each $D$ bin. Another difference in this database compared

to the ARTS particles, is that the snowflakes are now preferentially oriented so that their short dimension is in the vertical and longer dimensions are in the horizontal. As we will see later, this has some relevance for the extent of non-Rayleigh scattering at a given $D$.

By inspection of the simulation data we can estimate $\mathcal{A}$ directly from the region of the $IWC/Z$ curve where the values become nearly constant. In Fig. 3, we present simulations with the 7 different scattering models to show the effect of particle

habit. This figure follows the same format as Figs. 1f and 2f. Panels (a) and (b) show results using the ARTS mixtures, and panels (c) and (d) use particles from the Mroz and Leinonen (2023) database. The shading of the lines was chosen based on the value of $m_\lambda$ for each habit (see Table. 1). The particles with smallest $m_\lambda$ (i.e the unrimed dendritic aggregates) are plotted using the lightest shade of grey, with darker shades representing habits with increasing $m_\lambda$. It is clear that the different habits show similar qualitative behaviour, but different quantitative behaviour. The ratios are generally higher for lower mass

particles, and lower for more massive particles. This suggests that $m_\lambda$ is a key sensitivity for the method presented here. In other words, knowledge of $m_\lambda$ is required in order to determine the magnitude of the asymptotic value. We note that for a given $m_\lambda$, the ratios are different between the two databases. Since oriented particles have their mass distributed more widely in the horizontal and more narrowly in the vertical compared to particles with random orientation, a particle of a given mass will exhibit less non-Rayleigh scattering if it is oriented. This results in increased $c_f$, thus increasing $\kappa$, as seen in Table. 1.

Evidently, the curves are not perfectly flat. The plate and block aggregate mixtures from ARTS appear to approach an asymptotic limit more rapidly, whilst some of the others (such as the column aggregate mixture and the unrimed dendritic aggregates) show slight oscillations of $IWC/Z$ and $S/(Z \times \mathrm{MDV})$ with $D_m$. In the following section, we hypothesise that this is related to the form of the function $f$, which captures the deviation from Rayleigh scattering, and in particular how closely $f$ of the aggregate models follows the power law scaling assumed in Sect. 4.

Since, in practice, these curves are not perfectly flat, we choose a representative value at $D_m = 2\ \mathrm{mm}$ (i.e. centrally within the 1–3 mm interval used to estimate $c_{ns}$ and $c_{R_g}$, as discussed below) and tabulate this as $\mathcal{A}_{\mathrm{IWC}}$ in Table. 1.

For completeness, we also estimate the asymptotic value of $S/(Z \times \mathrm{MDV})$ which is tabulated as $\mathcal{A}_S$. Since our figures have $S$ in units of $\mathrm{mm\,h}^{-1}$, $\mathcal{A}_S$ and $\mathcal{A}_{IWC}$ are not numerically equal, but should be trivially related to each other by a unit conversion, and indeed we find these two independent estimates are consistent to within 6%.





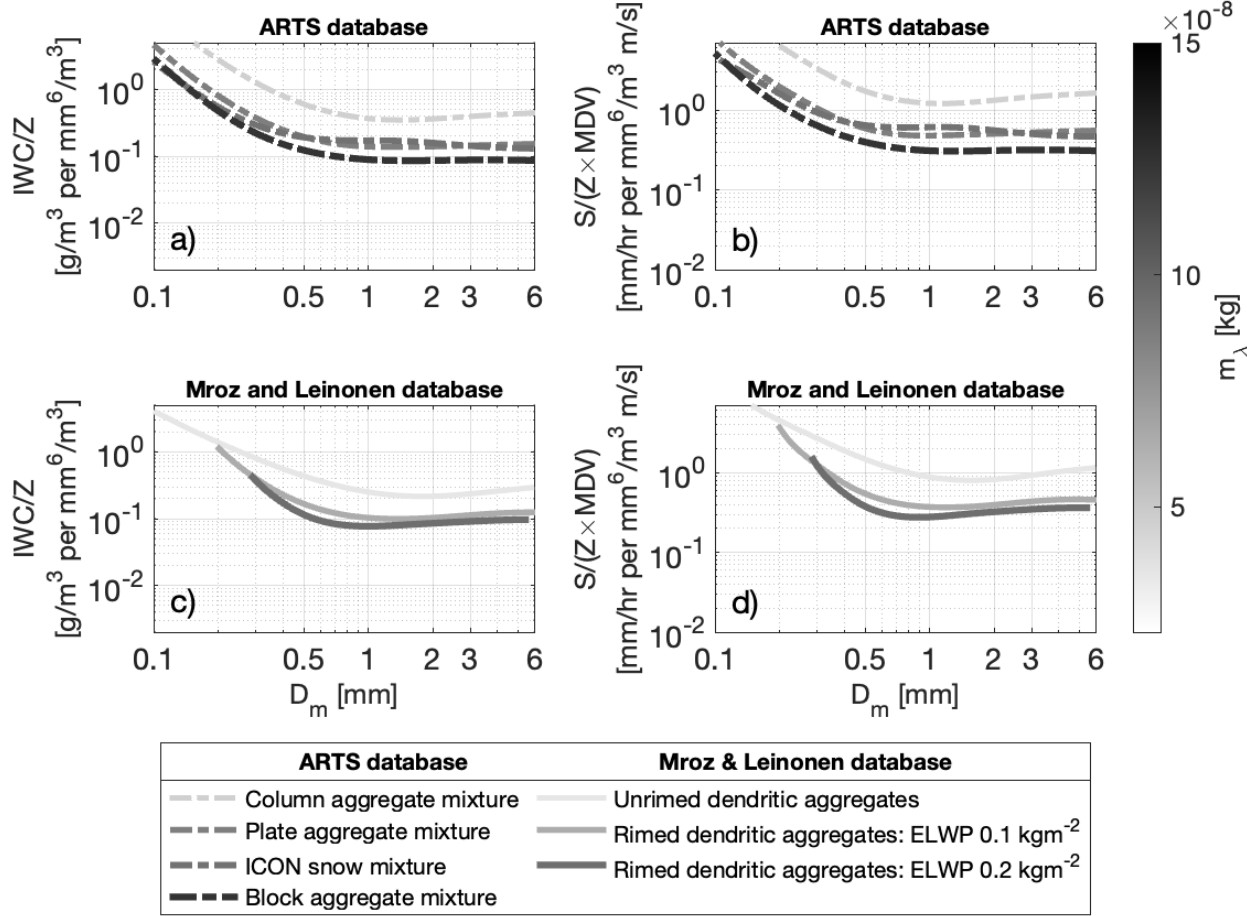

**Figure 3.** G-band simulations using exponential PSDs and different particle habits from the ARTS scattering database (Eriksson et al., 2018) (dashed lines in panels (a) and (b)) and the database of Mroz and Leinonen (2023) (solid lines in panels (c) and (d)). Panels (a) and (c) show the ratio IWC/Z, and panels (b) and (d) show $S/(Z \times \mathrm{MDV})$ for a range of $D_m$. The lines are shaded in order of the $m_\lambda$ values provided in Table. 1, with darker shades representing larger $m_\lambda$.

To work out the dimensionless non-spherical scattering coefficient $c_{ns}$, we calculated $Z$ at 3 GHz (i.e. in the Rayleigh regime) using an exponential PSD, and compared it to results calculated using Rayleigh spheres of the same mass (i.e. Eq. 7 with $c_{ns} = 1$). Dividing the two quantities gives $c_{ns}$. This was repeated for PSDs with different $D_m$, and an average of the values obtained in the interval $D_m = 1$–3 mm is shown in Table 1.

Estimation of $c_{R_g} = R_g/D$ is straightforward for the ARTS particles, since we have the full shape data of each particle, and from that we estimated $R_g$ and $D$ directly. Again, we computed a representative average value of this parameter. To do this, we first computed a mass-weighted average of $c_{R_g}^{-b}$ across the particle size distribution, and then inverted the result to obtain an



overall value of $c_{R_g}$ for the complete distribution:

$$c_{R_g,m} = \left[ \frac{\int_0^\infty N(D)m(D)c_{R_g}^{-b}dD}{IWC} \right]^{-1/b}.$$ (14)

Repeating this process for PSDs with $D_m = 1\text{--}3$ mm as before, we took a median of the resulting values and this is the figure shown in Table 1. The rationale for averaging $c_{R_g,m}^{-b}$ is that the coefficient appears in this (nonlinear) form in $\kappa$. $c_{R_g,m}^{-b} = \int_0^\infty N(D)m(D)c_{R_g}^{-b}dD/IWC$. As for $c_{ns}$, we use the median value for $D_m$ in the range 1–3 mm.

Since detailed shape data is not openly available for the particles from the database of Mroz and Leinonen (2023), $c_{R_g}$ cannot be calculated in the same way for those particles. However, Leinonen and Szyrmer (2015) suggest that $c_{R_g} \approx 0.287$, which is consistent with the values calculated for the ARTS particles, and this is the value we have used in our analysis.

After calculating $c_{ns}$ and $c_{R_g}$, the only unknown coefficient is $c_f$. To obtain $c_f$, we can use the asymptotic values of $\mathcal{A}$ from our simulations, and divide this by the value of $1/\kappa m_\lambda$ that would be obtained if $c_f$ were equal to one. We used $\mathcal{A}_{IWC}$ to determine the value of $c_f$ given in Table. 1. This means calculating $1/\kappa m_\lambda$ using $\kappa$ and $m_\lambda$ in the table is equivalent to $\mathcal{A}_{IWC}$ in the table (allowing for a unit conversion of $10^3$ to convert the SI units of IWC $\mathrm{kg\,m^{-3}}$ to the more convenient units of $\mathrm{g\,m^{-3}}$ used in Figs. 1 and 3).

The values of $c_{ns}$ and $c_{Rg}$ are quite consistent across the various particle models, with values lying in the ranges $1.16 \pm 6\%$ and $0.31 \pm 10\%$ respectively. There is a higher variation (factor of 3.2) in $c_f$. However the values are fairly consistent for the ARTS habits with values of $1.15 - 1.58$, while the majority of the variation in $c_f$ comes from the particles in the rimed database. A larger value of $c_f = 2.49$ is found for the unrimed dendritic aggregates, and the largest value of $c_f = 3.71$ is found for rimed dendritic aggregates with ELWP$= 0.1$ $\mathrm{kg\,m^{-2}}$. As riming increases to ELWP$= 0.2$ $\mathrm{kg\,m^{-2}}$, $c_f$ decreases to 1.92.

The coefficients are used to calculate $\kappa$ in Table. 1, which is found to be very similar for all four ARTS mixtures considered, varying by only 25%. However, $m_\lambda$ varies by a factor close to 4 for these mixtures. This implies that in some cases it may be suitable to assume a value for $\kappa$, meaning the only thing we need to know is $m_\lambda$. In other words, we don't need to assume a particular scattering model, the only parameter required to perform a retrieval is the mass of a wavelength-sized particle. There is greater variability in $\kappa$ for the particles from the rimed database. Considering all seven particle habits used here, $\kappa$ varies by a factor of 3 (which is likely to be driven by the large variation in $c_f$ since the variability is small for the other coefficients). However $m_\lambda$ varies more strongly, by a factor of 6.5. Thus we suggest that $m_\lambda$ is the primary sensitivity in controlling what $\mathcal{A}$ is.





**Table 1.** Relevant coefficients for the four particle mixtures from the ARTS scattering database (Eriksson et al., 2018) and the three habits from the rimed database of Mroz and Leinonen (2023) used in this study. * Since the shape data is not available for the particles from the database of Mroz and Leinonen (2023), the values of $c_{Rg}$ cannot be calculated directly. Thus we use the value of $c_{Rg} \approx 0.287$ suggested by Leinonen and Szyrmer (2015).

| | Units | Plate agg. mixture | Block agg. mixture | Column agg. mixture | ICON snow mixture | Dendritic aggs | Rimed dend. aggs. ELWP 0.1kgm$^{-2}$ | Rimed dend. aggs. ELWP 0.2kgm$^{-2}$ |
|---|---|---|---|---|---|---|---|---|
| $c_{ns}$ | - | 1.16 | 1.11 | 1.16 | 1.08 | 1.23 | 1.17 | 1.16 |
| $c_{Rg}$ | - | 0.28 | 0.29 | 0.28 | 0.34 | 0.287* | 0.287* | 0.287* |
| $c_f$ | - | 1.35 | 1.58 | 1.55 | 1.15 | 2.49 | 3.71 | 1.92 |
| $a$ | kgm$^{-b}$ | 0.21 | 0.35 | 0.25 | 0.031 | 0.0128 | 0.1847 | 0.1298 |
| $b$ | - | 2.26 | 2.27 | 2.43 | 1.95 | 2.035 | 2.288 | 2.154 |
| $A_{IWC}$ | $\frac{g/m^3}{mm^6/m^3}$ | $1.4 \times 10^{-1}$ | $0.9 \times 10^{-1}$ | $3.6 \times 10^{-1}$ | $1.6 \times 10^{-1}$ | $2.17 \times 10^{-1}$ | $1.03 \times 10^{-1}$ | $0.86 \times 10^{-1}$ |
| $A_S$ | $\frac{mm/hr}{(mm^6/m^3)(m/s)}$ | $5.1 \times 10^{-1}$ | $3.1 \times 10^{-1}$ | $13.4 \times 10^{-1}$ | $5.6 \times 10^{-1}$ | $8.2 \times 10^{-1}$ | $3.9 \times 10^{-1}$ | $3.2 \times 10^{-1}$ |
| $m_\lambda$ | kg | $9.58 \times 10^{-8}$ | $1.50 \times 10^{-7}$ | $3.80 \times 10^{-8}$ | $1.05 \times 10^{-7}$ | $2.30 \times 10^{-8}$ | $6.41 \times 10^{-8}$ | $1.08 \times 10^{-7}$ |
| $\kappa$ | mm$^6$kg$^{-2}$ | $7.47 \times 10^{10}$ | $7.74 \times 10^{10}$ | $7.24 \times 10^{10}$ | $6.01 \times 10^{10}$ | $18.49 \times 10^{10}$ | $13.81 \times 10^{10}$ | $9.95 \times 10^{10}$ |



## 6 Testing the assumed power-law scaling of the non-Rayleigh scattering ($f$)

As discussed in Sect. 4, Sorensen (2001) used RGA to demonstrate that for fractal aggregates large enough compared to the wavelength (i.e. $\frac{4\pi R_g}{\lambda} \gtrsim 1$), $f$ is characterised by the power law given in Eq. 10. We now test in more detail whether that approximation holds for realistic ice particle models from the scattering databases. The data in both databases were calculated using the numerically exact discrete dipole approximation. Provided the resolution of the discretisation is sufficient, the DDA provides an accurate solution to Maxwell's equations (Yurkin and Hoekstra, 2007).

The red lines in Fig. 4 show $f$ computed using Eq. 10, for the particle habits outlined in Table 1.

From the equations outlined in Sect. 4, we can write the non-Rayleigh radar cross section as:

$$\sigma_r = 36\pi^3 \lambda^{-4} \frac{m^2}{\rho_{ice}^2} \left| \frac{\epsilon - 1}{\epsilon + 2} \right|^2 c_{ns} f. \tag{15}$$

Thus we may also calculate $f$ for realistic ice particle models by using $\sigma_r$ and $m$ from the databases, and $c_{ns}$ from Table 1. The black markers in Fig. 4 show $f$ for various values of $\frac{4\pi R_g}{\lambda}$, computed for the various particles. In panels a-d, the values of $f$ for the monomers of each ARTS mixture are plotted using dots, while the crosses show $f$ for the aggregates of the mixture. At small values of $\frac{4\pi R_g}{\lambda}$, $f \approx 1$ and these particles correspond to the monomer habits in the mixture, along with some of the smaller aggregates. The crossover to power law scaling for $f$ begins at $\frac{4\pi R_g}{\lambda} \approx 1$ for each of the ARTS habits, following the expected behaviour of Sorensen (2001). Similar behaviour is seen for the particles from the Mroz and Leinonen (2023) database, but the crossover to power law scaling occurs at slightly larger values closer to 2. This shows that power law scaling is indeed evident in the scattering data for complex aggregates, rimed and unrimed, in the G-band, validating a cornerstone of our theoretical interpretation. Using the values of $c_{Rg}$ in Table 1, we can see that the point where the crossover to power law scaling begins corresponds to $D \approx \lambda/4$, at which size scattering from ice at the front and back of the particle would be out of phase in the backward direction. At 200 GHz ($\lambda = 1.5$ mm) this critical diameter is 0.375 mm.

As outlined in Sect. 4, the power law scaling of $f$ led to the results in equations 12 and 13, whereby an almost direct proportionality between $Z$ and IWC, and $Z \times$ MDV and $S$ was deduced. In other words, the power law scaling of $f$ drives the flattening of the ratios IWC/$Z$ and $S/(Z \times$ MDV) in the G-band. Although we determined in the previous section that $m_\lambda$ is the key parameter to determine the magnitude of the asymptotic value, the power-law behaviour of $f$ is the factor which drives the flattening of IWC/$Z$ in the first place. This influence is evident when comparing the data in Fig. 4 to Fig. 3. In Fig. 4, the data for the block and plate aggregate mixtures closely match the power law, and the ratios in Fig. 3 are the flattest for these habits. For example, in the range $D_m = 1$–6mm, the IWC/$Z$ ratio varies by only 3% and 11% for the blocks and plates, while there are larger, more systematic deviations from the idealised power law curve for the large column aggregate mixture and the ICON snow mixture, resulting in larger variations in IWC/$Z$ of 26% and 27%. The power law generally overestimates the column aggregate mixture data at larger values of $\frac{4\pi R_g}{\lambda}$. For the ICON snow mixture, the power law underestimates the data at $\frac{4\pi R_g}{\lambda} = 1.5$, overestimates it at $\frac{4\pi R_g}{\lambda} = 4.5$, and underestimates it again around $\frac{4\pi R_g}{\lambda} = 10$. Because $\lambda$ is fixed, and $R_g \propto D$, these under/overestimations feed through to the scattering properties of different particle sizes, and as $D_m$ is varied,



these particle sizes are weighted to a greater or lesser extent in the value of $Z$ and MDV, producing weak oscillations around
the anticipated constant value of IWC/$Z$ and $S/(Z \times \text{MDV})$.

## 7   Sensitivity to the form of the particle size distribution (PSD)

Analysis of in-situ observations has generally led to cloud PSDs being parameterised by either exponential or gamma distri-
butions e.g. Sekhon and Srivastava (1970); Heymsfield et al. (2002, 2013). In this subsection, we consider the effect of PSD
shape on our results. So far, we have assumed an exponential distribution shape. To investigate the sensitivity to this assump-
tion, G-band simulations for plate aggregates are repeated using gamma PSDs $N(D) = N_0 D^\mu \exp(-\Lambda D)$, where $\mu$ is a shape
parameter. If $\mu = 0$, the gamma PSD reduces to an exponential PSD, while increasing $\mu$ decreases the small particle concen-
tration, and produces a less disperse distribution. Negative values of $\mu$ are also possible, which give very broad distributions,
including large numbers of small particles. Care is required when $\mu < 0$ since some moments of the distribution can become
divergent - this is not the case for IWC, $S$ or $Z$ in the $\mu = -1$ case shown here. The different coloured lines in Fig. 5 show
results for representative values of $\mu$, ranging from $-1$ to $5$ (based on the analysis in Mason et al. 2019). Panels (a) and (b)
show the ratios IWC/$Z$ and $S/(Z \times \text{MDV})$ for a range of $D_m$. Furthermore, panels (c) and (d) show the percentage difference
of the ratios for nonzero $\mu$ values relative to our nominal simulations at $\mu = 0$ from Figs.1,2.

It is clear that in the region of $D_m$ we are interested in, the results are not very sensitive to the PSD shape parameter. Varying
$\mu$ between $-1$ and $2$ causes differences of less than 9% for IWC, and less than 6% for $S$, relative to using an exponential
distribution. The differences are higher, with more oscillatory behaviour, for the largest value of $\mu$ (5), but remain below 12%.
This weak oscillatory variation is likely because as $\mu$ increases, the PSD becomes increasingly narrow, and dominated by a
more restricted range of particle size. The scattering model comprises a single particle per size bin, and for this situation these
will be aggregates of varying (random) configurations. The scattering properties fluctuate with size as a result of the random
realisations, and the integrated values at high $\mu$ are more sensitive to these fluctuations, because of the narrower weighting
in $D$ space. In nature, these fluctuations may not be evident, because natural clouds probed by radar consist of an enormous
ensemble of particles at all sizes. We therefore suspect this sensitivity to $\mu$ is an upper limit to the true sensitivity to PSD shape.

At very small $D_m$ (below 0.3 mm) there is stronger sensitivity to the PSD shape, as the scattering becomes increasingly
influenced by smaller particles in the Rayleigh regime. However, since our target in this work is $D_m > 0.5$ mm, this is not a
concern.

## 8   Application to case studies

To illustrate the practicalities of retrieving IWC and $S$ using Eqs. 12 and 13, we now present measurements of two ice-phase
clouds which passed over the Chilbolton Observatory in the UK, using the GRaCE 200 GHz cloud radar. This instrument has
a 0.1° vertically-pointing beam, transmits around 80mW output power in a pulse of variable duration, and records full Doppler
spectra profiles, which can then be analysed to estimate $Z$ and MDV. Further details can be found in Courtier et al. (2022). The





**Figure 4.** Comparison of $f$ as a function of $\frac{4\pi R_g}{\lambda}$ calculated from data and theory for the different particle habits in this study. The black markers show $f$ calculated using Eq. 15. Panels a-d use data for the ARTS mixtures, where the dots and crosses represent that the particles are monomers and aggregates, respectively. Panels e-g use data for the aggregates from the rimed database. The red lines show the theoretical power law expression for $f$ shown in Eq. 10, illustrating that power law scaling is evident in the data at $\frac{4\pi R_g}{\lambda} \gtrsim 1$. Note for the ARTS particles $R_g$ is computed directly for each particle; for the Mroz and Leinonen (2023) database we assumed $R_g \approx 0.287D$, following Table 1.





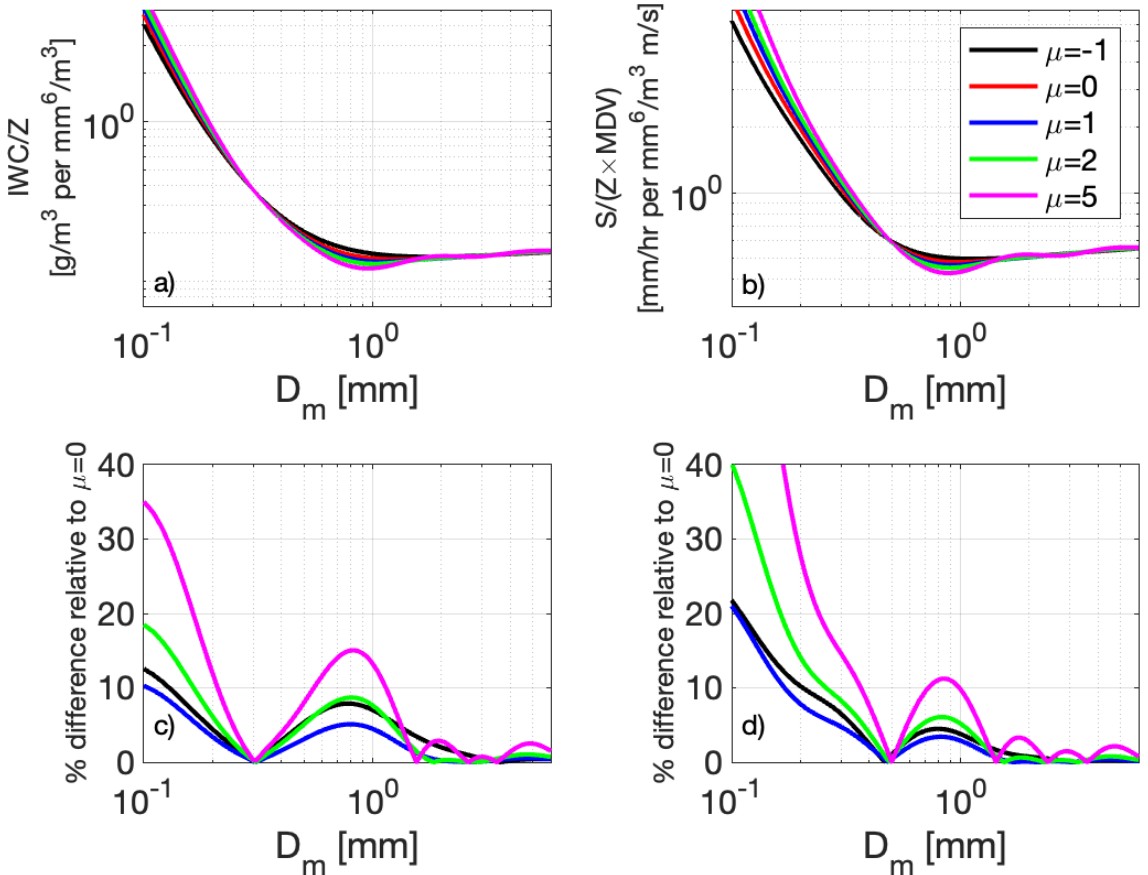

**Figure 5.** G-band simulations using the plate aggregate mixture and gamma PSDs. Panels (a) and (b) show the $\mathrm{IWC}/Z$ and $S/(Z \times \mathrm{MDV})$ ratios for a range of $D_m$. The different line colours correspond to different values of the shape parameter $\mu$, where $\mu = 0$ (red lines) is equivalent to an exponential PSD. Panels (c) and (d) show the percentage difference relative to the $\mu = 0$ case.

aim of this section is to (a) demonstrate how retrievals can be made in practice, and (b) to provide some in-situ evidence (from a precipitation gauge in Case 1, and aircraft measurements in Case 2) that the $S$ and IWC retrievals produced are realistic.

Attenuation is very important at G-band (Battaglia et al., 2014). In what follows, reflectivity data has been corrected for water vapour attenuation using the absorption model of Liebe (1989), along with nearby soundings at Larkhill (approximately 30 km from Chilbolton; Case 1) and a dropsonde released at 11:22 ($<$ 10 km from Chilbolton; Case 2). The vapour attenuation profiles are shown in Fig. 6.

Attenuation by ice particles in the cloud is much smaller than from water vapour, but not completely negligible. Fig. 7a shows a relationship between attenuation in the G-band and reflectivity at Ka-band, derived from simulations using the particle





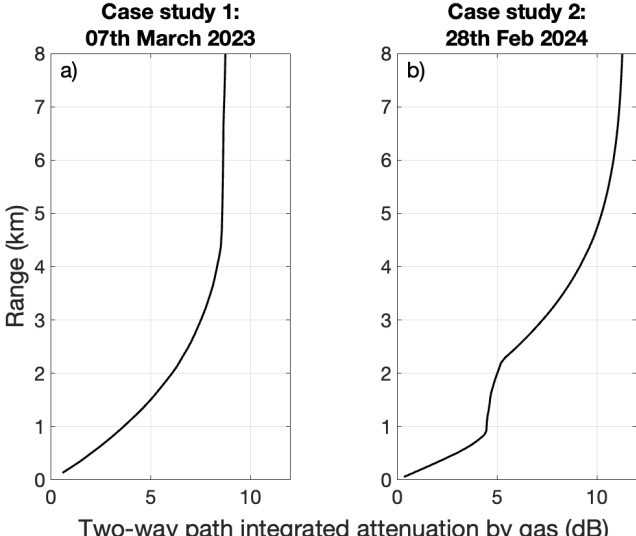

**Figure 6.** Two-way path integrated attenuation by water vapour in the G-band calculated using the absorption model of Liebe (1989) and soundings from Larkhill on (a) 7 March 2023, and (b) 28 February 2024.

models in this study (see Appendix A). The advantages of using the Ka-band reflectivity as a variable to diagnose attenuation

is the fact that there is significantly less attenuation from all sources (vapour, ice, liquid water) than there is in the higher frequency bands, so the value of $Z$ that is measured at Ka-band is the true reflectivity of the particles at that frequency. In the cases presented here, Ka-band data from the Copernicus 35 GHz cloud radar with a 0.25° vertically-pointing beam was also available (Walden, 2025). This allows us to diagnose attenuation by ice using the relationship shown in Fig. 7a, which is used to correct the G-band reflectivity data. In Case 1, the estimated two-way path integrated attenuation by ice at 4 km ranges

from values $< 0.1$ dB to about 4.8 dB, with an average correction throughout the day of around 1.4 dB. In Case 2, the path integrated attenuation at 8 km is less variable throughout the day, ranging from 0.6–2.6 dB, while the average correction is similar to Case 1, at 1.3 dB.

Since the ideas presented in this manuscript could, in principle, be applied when only a single frequency is available, Fig. 7b shows the same relationship but now correlating attenuation at G-band with *unattenuated* reflectivity at G-band. Application

of this second relationship is more complicated, because it requires the unattenuated reflectivity to be estimated. One approach might be to correct the reflectivity from the surface upward, moving range gate by gate. We include this information for completeness, but in what follows we will apply ice attenuation corrections based on the Ka-band profile measured at the same time.

Supercooled liquid water is also a significant source of attenuation at G-band (Battaglia et al., 2014). In Case 2 there is no

evidence of significant liquid water. In Case 1, retrievals from a collocated RPG HATPRO-G5 microwave radiometer (Humidity And Temperature PROfiler; Walden (2024a)), using the instrument's software, suggest that the liquid water path is typically





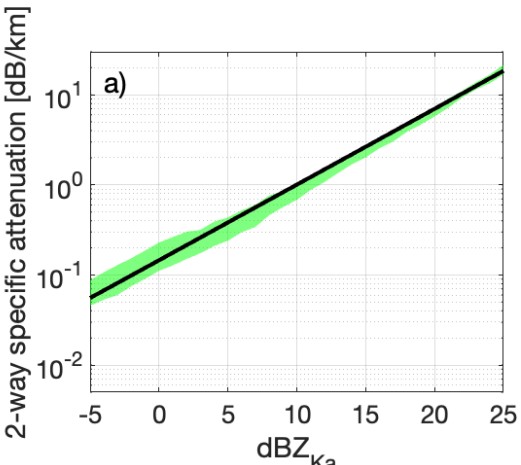 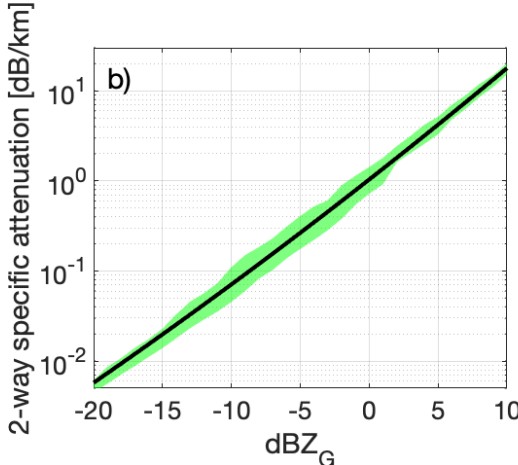

**Figure 7.** Panel (a) shows the relationship derived between $dBZ_{Ka}$ and attenuation by ice in the G-band. We used this relationship to correct for attenuation by ice in the data presented here. Panel (b) shows a fit derived in the same way, but with $dBZ_G$ on the x-axis. Note that the x-axis limits in (a) and (b) are different. The green shaded regions show the interquartile range of the data, representing variations due to differences in the scattering model and PSDs.

between 50 to 200 $\mathrm{g\,m^{-2}}$ (Fig. 8c). This corresponds to a total 2-way attenuation through the cloud of approximately 1 to 4 $\mathrm{dB}$ in the G-band. In what follows we do not attempt to correct for this, which may lead to underestimation of IWC and $S$ in the upper portions of the cloud system. However, the key data of interest for our analysis is in the lower parts of the cloud (near 365    1 $\mathrm{km}$ height), where the attenuation is likely to be smaller, and the agreement with gauge data in Case 1 suggests that this is not a major effect, particularly given the uncertainties inherent in comparing gauge data from the surface with radar data aloft.

## 8.1   Case study 1: 7 March 2023

On 7 March 2023 a shallow snow band passed over the observatory, and was sampled by the GRaCE radar. This case study is analysed in more detail in McCusker et al. (in prep.), which includes a dual-frequency retrieval of $D_m$, indicating that almost 370    everywhere in the cloud field has $D_m > 0.5$ mm. This is therefore an appropriate target for our retrieval method.

Panels (a) and (b) of Fig. 8 show dBZ and MDV measured by the GRaCE G-band radar over a 4 h period. During this time both the cloud depth (4 $\mathrm{km}$) and liquid water path ($\sim 100\ \mathrm{g\,m^{-2}}$) were reasonably consistent, albeit with some localised fluctuations in both quantities. After 14:00 UTC the cloud became increasingly shallow and broken, with larger, more variable liquid water path. During the 4 h window presented here, the majority of the reflectivity values lie in the range -10 to 0 $\mathrm{dBZ}$. 375    Doppler velocities are typically around $0.3\ \mathrm{m\,s^{-1}}$ at 3 $\mathrm{km}$ (positive velocity is downward), increasing in magnitude at lower altitudes to around $0.7$–$1.5\ \mathrm{m\,s^{-1}}$ at 1 $\mathrm{km}$.

There were no airborne in-situ cloud measurements collected on this day, so cloud particle imagery is not available to constrain the choice of scattering model. However, McCusker et al. (in prep.) examined the Doppler spectra for this case study.





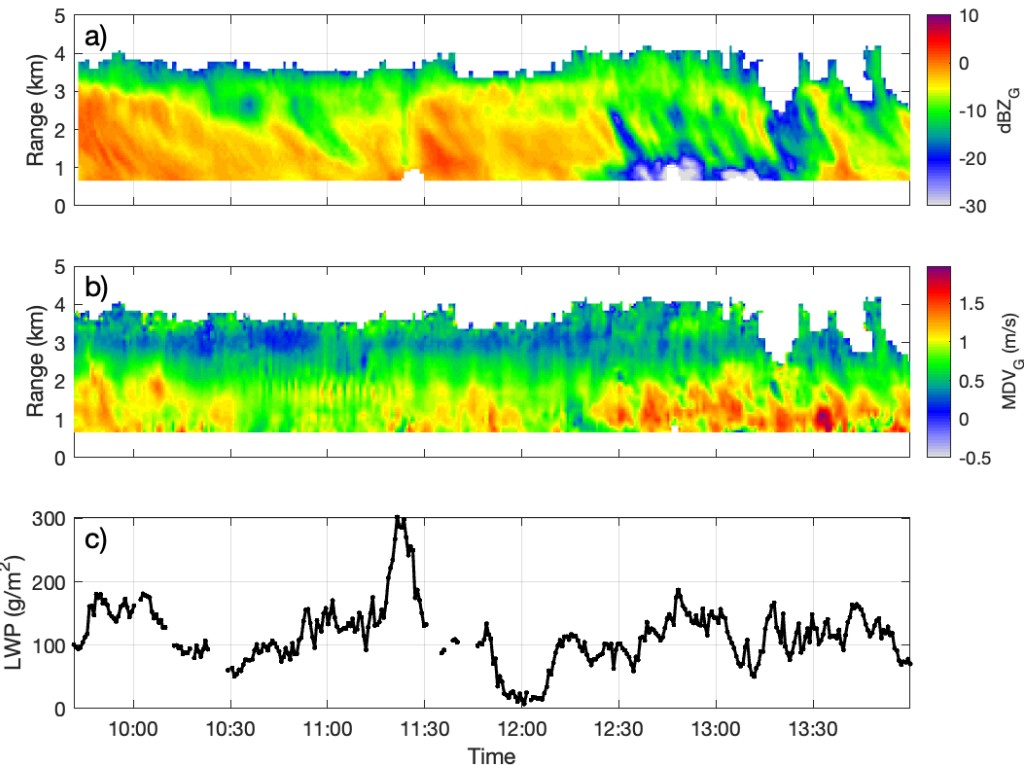

**Figure 8.** Panels (a) and (b) show dBZ and MDV measured by the GRaCE G-band radar at Chilbolton observatory on 7 March 2023. The liquid water path (LWP) retrieved from a collocated RPG HATPRO-G5 microwave radiometer (Humidity And Temperature PROfiler; Walden (2024a)), using the instrument's software, is shown in panel (c).

Through analysis of triple frequency diagrams in spectral space, they concluded that the measurements were consistent with
the Mroz and Leinonen (2023) rimed snowflakes. Taking this as guidance, and acknowledging that in Fig. 8c the liquid water path (LWP) oscillates around 0.1 $\mathrm{kgm^{-2}}$, we use the rimed dendritic aggregates with ELWP 0.1 $\mathrm{kgm^{-2}}$ from Table. 1 for the retrieval. Using these values we simply compute $IWC = Z \times \mathcal{A}_{IWC}$ and $S = Z \times MDV \times \mathcal{A}_S$.

The resulting IWC and $S$ values are shown in panels (a) and (b) of Fig. 9. Under the assumption of rimed dendritic aggregates, the retrieved IWC is typically a few hundredths of a gram per cubic metre, with a few isolated fallstreaks of larger IWC
values reaching 0.17 $\mathrm{gm^{-3}}$. Light snowfall rates with values of $S$ generally less than 0.3 $\mathrm{mm\,h^{-1}}$ are retrieved, with some isolated streaks producing higher values around 0.6 $\mathrm{mm\,h^{-1}}$.

To assess the realism of the retrieved fields, we have compared the snowfall rate retrieved in the lower parts of the cloud (we use the lowest $\sim$500 m of available data) to a fast response precipitation gauge at the ground (Norbury and White, 1971). The sounding from Larkhill indicates that the 0° C wet-bulb isotherm is around 100 m above the surface on this day, so the





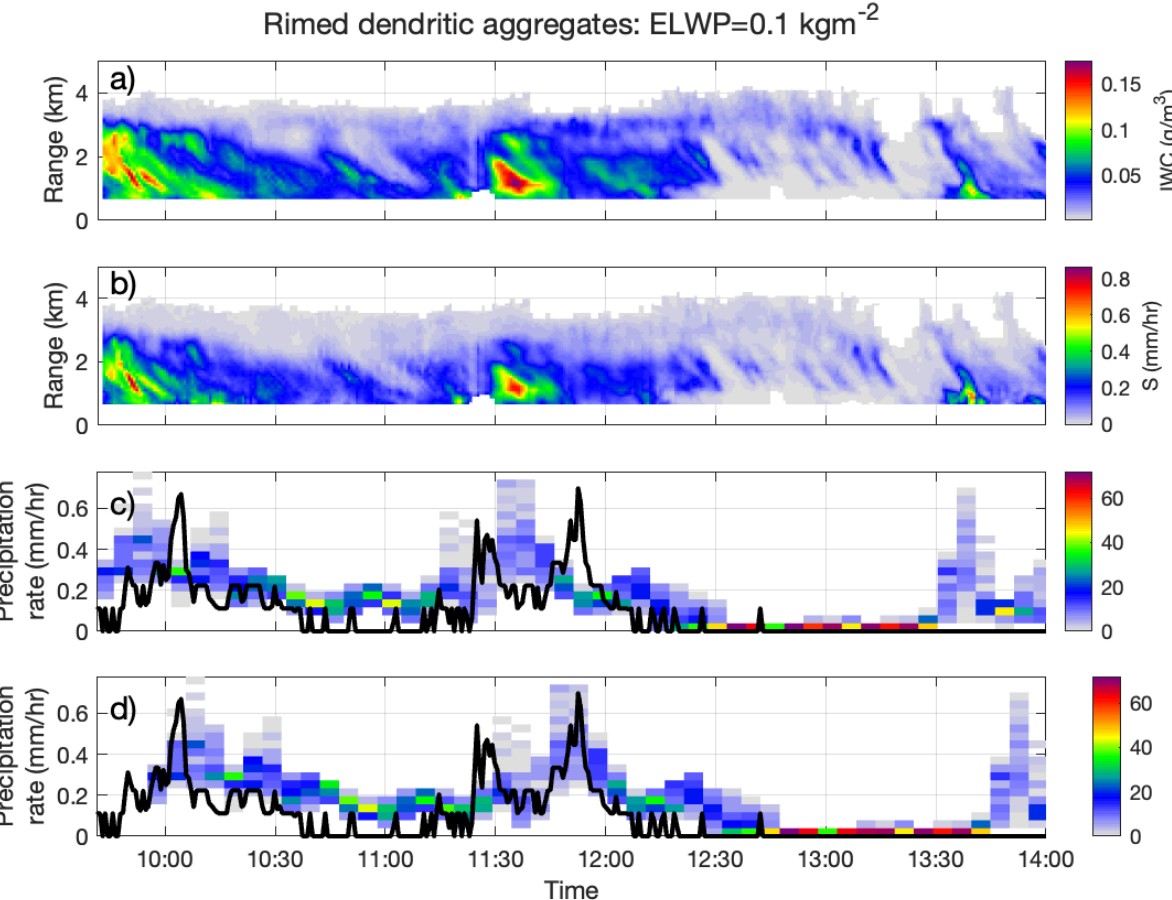

**Figure 9.** Panels (a) and (b) show IWC and $S$ retrieved from the measured $Z$ and MDV on 7 March 2023 shown in Fig. 8 using $\mathcal{A}_{IWC}$ and $\mathcal{A}_S$ estimated from the simulations for rimed dendritic aggregates with ELWP=0.1 kgm$^{-2}$. The coloured histogram data in panels (c) and (d) shows the values of retrieved $S$ from the bottom $\sim$500 m of cloud. Overlaid on the histograms is a black line showing the one-minute moving average precipitation rate measured from a drop counting rain gauge at Chilbolton. In panel (d) the histogram data has been shifted in time by 14.5 minutes to account for the approximate time it may take for precipitation to reach the rain gauge at the ground.

snowflakes are likely to have at least partially melted by the time they reach the gauge funnel. The results are shown in panels (c) and (d). The colours are time series of the histogram of $S$ retrieved from the radar aloft over the 500 m deep layer. The black line in each panel displays the precipitation rate from the gauge, showing measurements of light precipitation. In panel (c) it is clear that peaks in the precipitation rate at the ground are measured at a later time than when peaks in $S$ are retrieved in the lower cloud regions. This corresponds to the time taken for particles in this region to reach the ground. In panel (d)

we account for a time delay by shifting the histogram data in time by 14.5 mins. This corresponds to particles in the 500 m deep layer falling at approximately 0.8–1.3 ms$^{-1}$. Qualitatively, the histogram of retrieved $S$ displays similar behaviour to the





rain gauge measurements. At times of higher precipitation rate, the location and magnitude of the peaks are captured quite accurately in the retrieval. At times of low precipitation rate, the values of retrieved $S$ are slightly larger than observed by the gauge. This could be the result of evaporation, or variations in the properties of the particles in time (for example the mass-size relationship may change as the amount of riming varies). Towards the end of the time series, there is a peak in retrieved $S$ that is not reflected in the gauge measurements. This discrepancy is likely due to the narrow, small-scale nature of the feature, as evident in panels (a) and (b) of Fig. 8. The retrieval represents conditions approximately 1 km above the surface, and we are assuming vertical precipitation when performing a comparison with measurements at the surface. However, in reality, horizontal advection or wind drift may cause the precipitation to fall outside the gauge's catchment area, leading to a lower recorded value at the surface. This highlights the strength of radar-derived snowfall rates in resolving localised precipitation that surface gauges may fail to detect.

To assess the sensitivity of our results to the choice of particle model, we repeat the analysis from Fig. 9 using different scattering assumptions (Fig. 10). Panel (b) shows the original result obtained using rimed dendritic aggregates with an ELWP of 0.1 $\mathrm{kgm}^{-2}$. For comparison, panel (a) uses unrimed dendritic aggregates (i.e., lower-mass particles), while panel (c) uses rimed aggregates with a higher ELWP of 0.2 $\mathrm{kgm}^{-2}$. The use of unrimed particles results in larger retrieved values of $S$, approximately doubling compared to the baseline in panel (b). In contrast, increasing the riming from ELWP = 0.1 to 0.2 $\mathrm{kgm}^{-2}$ yields only a modest reduction in $S$ of about 0.1 $\mathrm{mm\,h}^{-1}$, corresponding to a difference of roughly 20% between the two rimed models. These results reflect the differences in $\mathcal{A}_S$ values in Table. 1, which vary more substantially between the unrimed and rimed models, but remain similar for the two rimed models.

## 8.2 Case study 2: 28th February 2024

The GRaCE G-band radar also collected measurements from Chilbolton on 28th February 2024. The Facility for Airborne Atmospheric Measurements (FAAM) BAe-146 research aircraft (http://www.faam.ac.uk) performed a flight on this day (C374), operating as part of the Characterising CirRus and icE cloud acrosS the specTrum-Microwave (CCREST-M) project. This provided a unique opportunity to collect G-band radar measurements from GRaCE along with sampling in-situ microphysics almost coincidentally.

GRaCE collected observations of cloud associated with an approaching warm front. The system eventually brought rain which attenuated the G-band signal. We focus on the deep stratiform ice cloud ahead of the surface rainfall. Panels (a) and (b) of Fig. 11 show dBZ and MDV from 09:12−11:48 UTC. Lidar data collected from the Vaisala CL51 instrument (Walden, 2024c) during this case study revealed some low-level liquid cloud beneath the ice cloud, at a height of approximately 0.3–0.6 km. The total 2-way attenuation by the liquid cloud was calculated using the LWP retrieved from the RPG HATPRO-G5 microwave radiometer using the MWRpy software (Walden (2024b); Fig. 11e), and the reflectivity values from the ice cloud have been corrected based on those estimates. Similar to the first case study, the reflectivity values in panel (a) typically range from around −10 to 0 dBZ. As well as a systematic trend for higher reflectivities at lower altitude, fall streaks are also apparent in the data indicating significant horizontal variability within the cloud. Doppler velocities mainly range from $\approx 0.5$ to 1.2 $\mathrm{m\,s}^{-1}$.





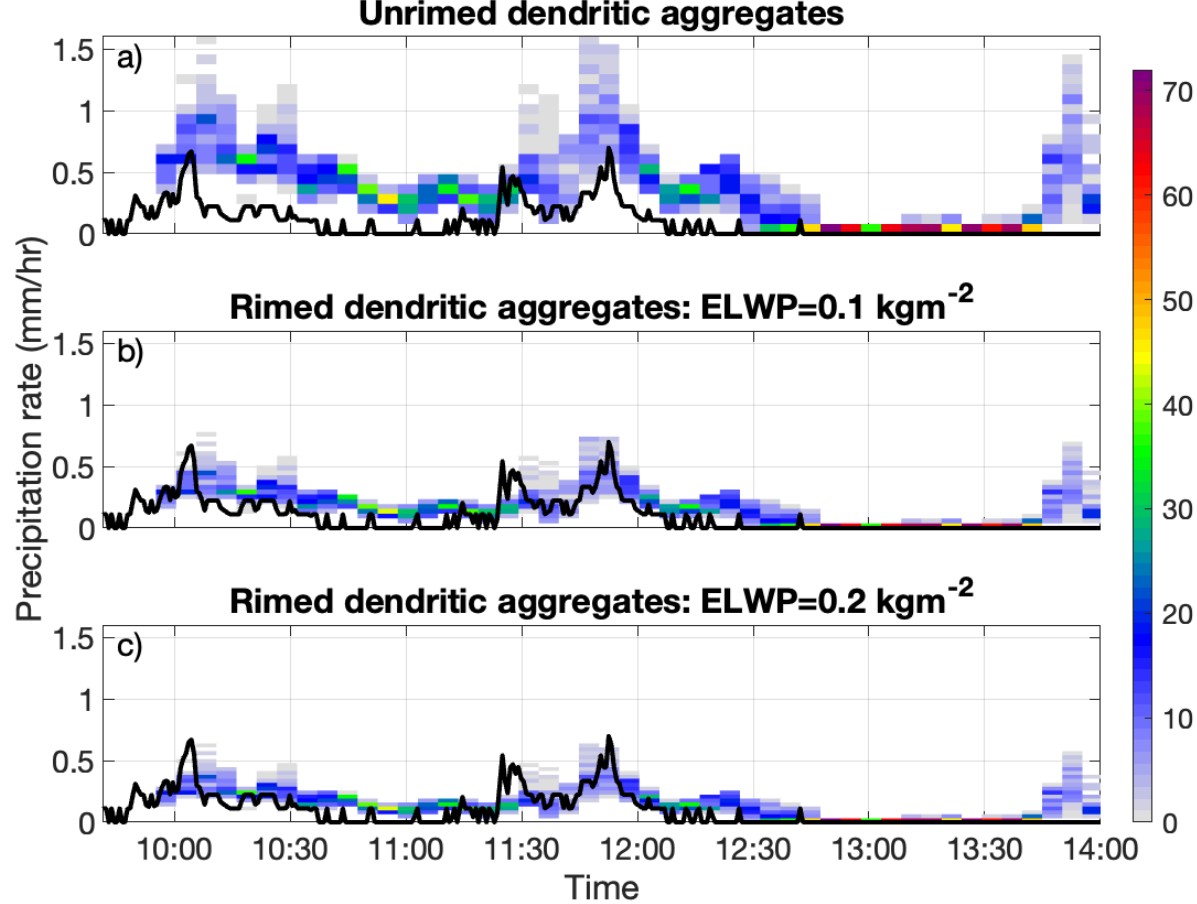

**Figure 10.** Sensitivity of $S$ retrievals to the choice of particle habit. As in Fig. 9d, the histograms show $S$ retrieved from the bottom $\sim$500 m of cloud, while the black lines show the one-minute moving average precipitation rate measured from a drop counting rain gauge at Chilbolton. Panels a-c show results using the three particle habits from the database of Mroz and Leinonen (2023). Panel (a) uses unrimed dendritic aggregates, panel (b) uses rimed dendritic aggregates with ELWP=0.1 kgm$^{-2}$ (i.e. the original result in Fig. 9d), and panel (c) uses rimed dendritic aggregates with ELWP=0.2 kgm$^{-2}$.

As discussed in previous sections, $m_\lambda$ is the key sensitivity for the method presented here. From Table. 1, it can be seen that for the ARTS aggregate mixtures, $\kappa \approx (7 \pm 1) \times 10^{10}$ mm$^6$kg$^{-2}$. Thus, in many cases where the cloud particles are mixtures of unrimed pristine crystals and aggregates, one could set $\kappa = 7 \times 10^{10}$ mm$^6$kg$^{-2}$ and calculate $m_\lambda$ using a pre-defined mass-size relationship, such as the one given by Brown and Francis (1995). This eliminates the requirement of a specific particle habit assumption. Since the particle imagery shows that the dominant crystal habits throughout this case study were unrimed bullet

rosettes and columns, along with relatively small aggregates of those habits (see CIP-15 imagery in Fig. 12), we believe this is a suitable method to use here. Using the Brown and Francis (1995) mass-size relationship corresponding to $D$ (as presented in





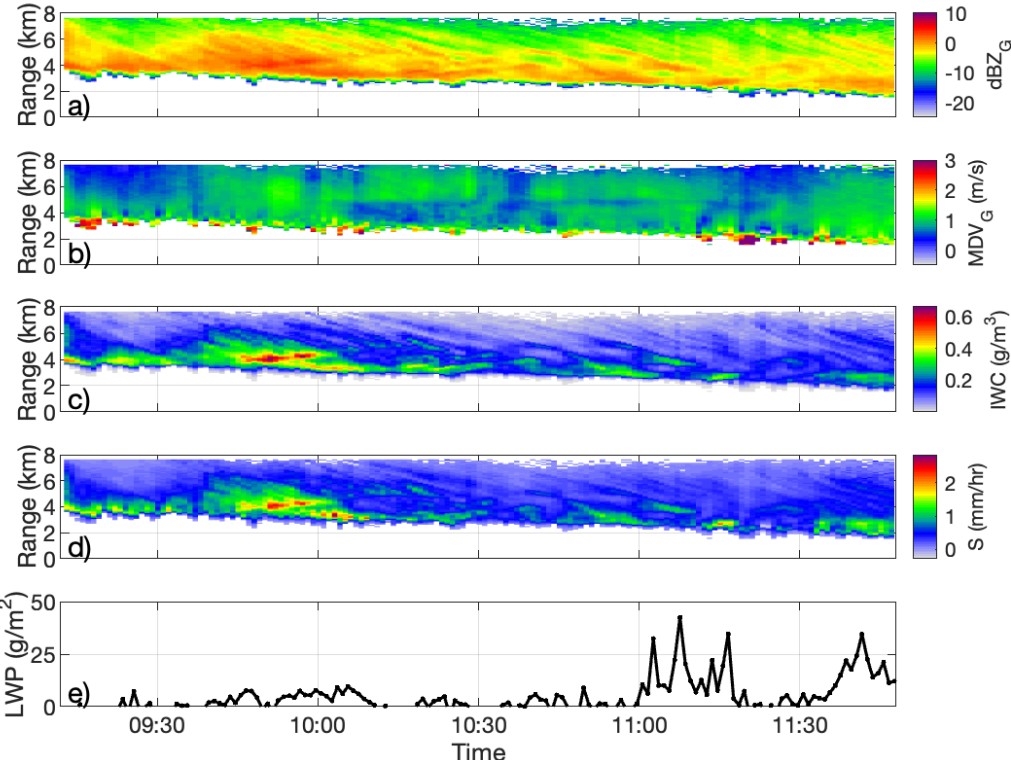

**Figure 11.** Panels (a) and (b) show dBZ and MDV measured by the GRaCE G-band radar at Chilbolton observatory on 28 February 2024, while panels (c) and (d) show retrieved IWC and $S$. The retrievals assume that $\kappa = 7 \times 10^{10}$ mm$^6$kg$^{-2}$, and $m_\lambda$ is calculated using the Brown and Francis (1995) mass-size relationship. Panel (e) shows the LWP retrieved from the RPG HATPRO-G5 microwave radiometer, using the MWRpy software (Walden, 2024b).

Hogan et al. (2012)) results in a value of $\mathcal{A} = 1/\kappa m_\lambda = 2.8 \times 10^{-1}$ g mm$^{-6}$. This value of $\mathcal{A}$ is used to convert measurements of $Z$ and MDV to IWC and $S$ ($IWC[\text{gm}^{-3}] = Z \times \mathcal{A}$; $S[\text{mm h}^{-1}] = Z \times \text{MDV} \times \mathcal{A} \times 3.6$). These are shown in panels (c) and (d) of Fig. 11. The retrieved ice water content varies from a few hundredths of a gram per cubic metre up to around $0.6$ g m$^{-3}$
in certain regions of the cloud (4.5 km, 09:50 UTC). The snowfall rate has a similar dynamic range, with rates of a few tenths of a millimetre per hour up to a peak of $\approx 2.5$ mm h$^{-1}$.

In Fig. 13 we plot the IWC estimates from the G-band radar as a 2D probability histogram. The black line overlaid on the histogram shows a profile of IWC estimated using measurements of the liquid and total water (ice plus liquid) contents from the Nevzorov probe, which is accurate to within about 0.002 g m$^{-3}$ (see Abel et al. (2014) for a description of the probe on
the FAAM BAe-146 research aircraft). The aircraft performed a stepped descent through the cloud and we use data collected between $11:18-12:48$ UTC in order to obtain an IWC profile throughout the entire height range. The profile was constructed



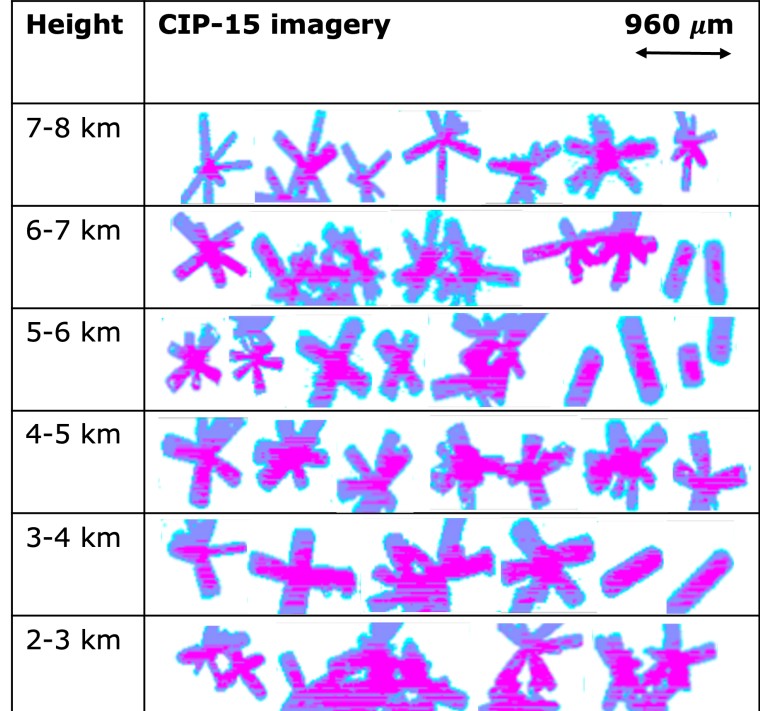

**Figure 12.** Examples of CIP-15 imagery at different heights on 28 February 2024. The pixel resolution is 15 $\mu$m.

by averaging Nevzorov IWC measurements collected at different altitudes during this time into 90 m range bins to match the range resolution of GRaCE on that day. The grey shaded region shows the range of values measured in each 90 m altitude bin, with the differences resulting from cloud inhomogenity. The red line overlaid on the histogram shows a profile of IWC obtained from integrating the PSD measured with the CIP-15 and CIP-100 probes during the aircraft descent, assuming the Brown and Francis (1995) mass-size relationship. As with the Nevzorov data, the shading indicates the range of values within each bin. This shows that the Brown and Francis (1995) relationship is generally consistent with the aircraft measurements. However, between about 4.8-6 km, the CIP IWC is lower than the Nevzorov IWC, indicating that particles in this region may have a higher mass than what is predicted by the Brown and Francis (1995) relationship.

We also estimated $D_m$ using the CIP PSD data, which was found to be $> 0.5$ mm at ranges of 7 km and below. Thus we expect the retrieval to be applicable to the majority of this cloud. $D_m < 0.5$ mm at the very top of the cloud (higher than 7 km), so we may expect the retrieval to have lower accuracy in this region.

This comparison allows us to test the realism of the retrieval using the in-situ data, but the colocation of the two datasets in time and space is imperfect since the radar data shown were collected during the time period $09{:}12-11{:}48$ UTC, while the IWC data were collected between $11{:}18-12{:}48$ UTC. Moreover the radar was sampling vertical profiles at Chilbolton, while the aircraft sampled along a 120 km long radial extended out to the southwest of Chilbolton. As a result, only a statistical comparison is possible. Fig. 13 shows that the Nevzorov IWC profile falls within the distribution of the retrieved values (with



the exception of a shallow layer near 4.5 km height). Above 5 km the aircraft profile follows the peak of the radar distribution to within $\approx 0.03\,\mathrm{g\,m^{-3}}$. At lower altitudes the IWC profile from the aircraft shows large non-monotonic variations with height,

which suggests there is significant inhomogeneity in the cloud. Despite these large fluctuations, the data still appear to fluctuate around the centre of the distribution of the radar retrievals. Overall, this comparison gives us further confidence that the method we are investigating in this paper produces realistic results.

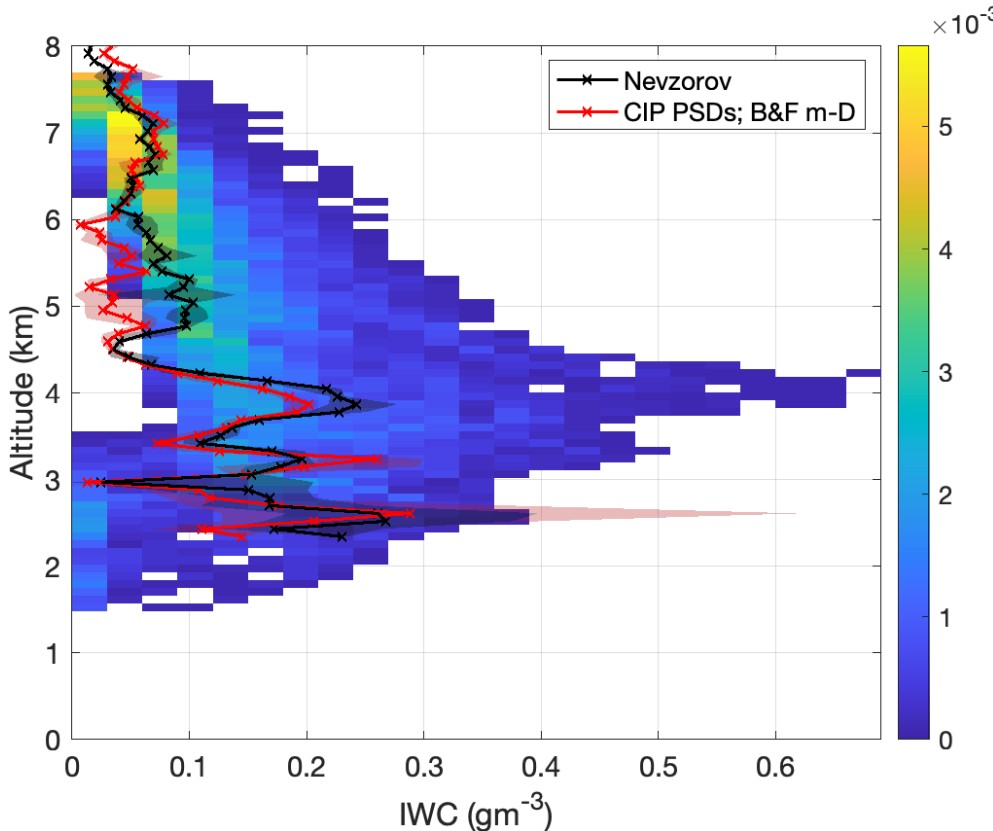

**Figure 13.** Probability histogram of retrieved IWC from dBZ$_G$, using $\kappa = 7 \times 10^{10}\ \mathrm{mm^6 kg^{-2}}$ and $m_\lambda$ from the mass-size relationship of Brown and Francis (1995). The black line shows the measured IWC profile using the Nevzorov probe on board the FAAM aircraft. The red line shows the IWC obtained from integrating the PSD measured with the CIP probes, assuming the Brown and Francis (1995) mass-size relationship. Both IWC profiles are averaged into 90 m range bins to match the range resolution of GRaCE, but shaded regions are included to show the range of values measured in each altitude bin.



## 9   Discussion and conclusions

In this study, we demonstrate, for the first time, the near-linear relationship between G-band radar reflectivity and ice water
content, leveraging the strong non-Rayleigh scattering characteristics unique to these frequencies. This simplifies retrieval
methods significantly compared to lower-frequency radars. We explore the strong non-Rayleigh scattering produced by ice
particles in the G-band, and present theory and simulations to show that measurements of $Z$ and $Z \times \mathrm{MDV}$ are almost directly
proportional to IWC and $S$, respectively. This is in stark contrast to the behaviour that occurs at lower frequencies, and presents
the opportunity for simple but accurate retrievals of cloud ice quantities. In principle, these retrievals could take place using
G-band measurements alone, provided the profiles can be corrected for attenuation by water vapour (the dominant component),
liquid water (if present) and ice. Alternatively the approach here can be incorporated into a more sophisticated multi-frequency
retrieval. One approach would be to use the simple $Z - \mathrm{IWC}$ relationship as a robust first estimate of the water content profile,
which is subsequently refined using the dual frequency ratio data.

We provide theoretical background and analysis to show that the near constant values of $\mathrm{IWC}/Z$ and $S/(Z \times \mathrm{MDV})$ are
expected for fractal aggregates that are large compared to the wavelength. The behaviour is driven by the power law scaling of
$f$, the dimensionless factor used to express non-Rayleigh scattering behaviour, and is consistent with the analysis of Sorensen
(2001). Since this latter behaviour was originally derived using RGA (which, as McCusker et al. 2019 and others have shown,
does not capture the full physics of the scattering process), and since radar measurements of ice particles were not the intended
application of Sorensen's study, it is necessary to turn to DDA scattering data on realistic ice particles to evaluate whether
the anticipated scaling holds for real snowflakes. The simulations show that the power law scaling is indeed evident, and
the expected linear relationship between IWC and $Z$ is obtained. Some small fluctuations around this linear relationship are
present (which are more evident when visualising their ratio as a function of $D_m$ in figures 1, 3). Digging into the details
on the non-Rayleigh function $f$ indicates that this fluctuation is driven by deviations from the overall power law scaling of
$f$. We speculate that these deviations are driven by variability in the particle geometry (e.g. different random realisations of
aggregates) for different particle sizes. The direct proportionality of IWC and $Z$ is most accurate when $f$ of the assumed
particle model closely follows this assumed power law scaling.

The expected magnitude, $\mathcal{A}$, of the asymptotic value of $\mathrm{IWC}/Z$ is dependent on $\kappa$ and $m_\lambda$, but crucially is *not* sensitive to
PSD parameters, and we show that the dominant factor controlling the magnitude of $\mathcal{A}$ is $m_\lambda$, i.e. the mass of a wavelength-
sized ice particle. If $m_\lambda$ is known (i.e. we can place a constraint on the mass-size relationship), then we expect retrievals to have
uncertainties within $\approx \pm 30\%$ for IWC, and within $\pm 15\%$ for $S$. These uncertainties become even narrower for $D_m \gtrsim 0.8\,\mathrm{mm}$.
The variability of $\kappa$ between different particle scattering models is significantly smaller than the variability of $m_\lambda$, varying by
only 25% for the four unrimed particle mixtures from the ARTS database. In contrast, $m_\lambda$ varies by a factor of almost 4
between the ARTS models. This suggests that it may be acceptable to fix the value of $\kappa$, leaving $m_\lambda$ as the only free parameter
to select in the retrieval method. This in turn would mean there is no requirement to base the retrieval on the existing scattering
database particle models - instead it is sufficient to know $m_\lambda$ from an appropriate mass-size relationship. We note that the
unrimed and rimed dendritic aggregates from the database of Mroz and Leinonen (2023) have larger values of $\kappa$, which we



propose is caused by the increase in $c_f$ resulting from particle orientation. However, $\kappa$ for all seven particle models considered here varies by a factor of 3, which is still considerably less than the factor of 6.5 variability in $m_\lambda$. In the future, we plan to examine the behaviour of different particle models and test the overall variability in $\kappa$ and other parameters. Such an analysis
would benefit from the availability of additional scattering data in the G-band.

The method is applicable in the regime where particles dominating the radar measurements are comparable to or larger than a quarter of a wavelength, and our data indicates this is satisfied for clouds where $D_m \gtrsim 0.5$ mm. Particles in this size range are frequent in low and mid-level ice clouds where $0 < T < -20^\circ C$ (Field et al., 2005). In high level cirrus clouds, the ice particles may be smaller than this, and the errors incurred in applying this method will be larger. The decline in accuracy for
small particles is gradual and monotonic: at $D_m = 0.35$ mm, $\mathrm{IWC}/Z$ and $S/(Z \times \mathrm{MDV})$ are typically a factor of 2 larger than the values in Table 1. At $D_m = 0.2$ mm, $\mathrm{IWC}/Z$ is typically a factor of 6 larger than the values in Table 1, while $S/(Z \times \mathrm{MDV})$ is typically a factor of 4 larger. These deviations lead to underestimates of IWC and $S$. In addition, we have shown in Fig. 5 that there is increased sensitivity to the shape of the PSD at low $D_m$.

We applied the theory to two case studies, showing that the method gives feasible results. Firstly, we explored a case study
from 7 March 2023 and compared values of retrieved $S$ at low altitude to the precipitation rate measured at the ground, which showed similar results and a well correlated time series. Secondly, we looked at a case study from 28 February 2024. The availability of in-situ measurements on this day allowed comparison of the retrieved IWC to measurements made using the Nevzorov probe on board the FAAM aircraft. This comparison showed that the measurements generally fall within the distribution of the retrievals. In this latter retrieval, $\mathcal{A}$ was calculated using $\kappa = 7 \times 10^{10}$ mm$^6$kg$^{-2}$ and $m_\lambda$ using the mass-
size relationship of Brown and Francis (1995), thus eliminating the requirement of a specific particle habit from a scattering database. Further in-situ validation experiments involving coincident radar measurements and aircraft-based particle sampling would significantly strengthen confidence in the retrieval method.

We have shown that the ratios $\mathrm{IWC}/Z$ and $S/(Z \times \mathrm{MDV})$ become flatter with increased frequency as one moves from Ka to W to G-band, and the value of $D_m$ above which the ratios become almost constant also decreases at higher frequencies. In
order for a direct retrieval method to be accurate at even smaller sizes ($D_m < 0.5$ mm), radars operating at frequencies higher than 200 GHz could be considered. This may allow the retrieval of IWC in high level cirrus clouds where the particles are typically smaller (Field et al., 2005). The trade-off is that gaseous attenuation becomes very large in the lower atmosphere at sub-millimetre wavelengths. However, this may be acceptable if the aim is to target the upper parts of the troposphere from space or airborne platforms, since the density of water vapour is much smaller, and therefore the gas attenuation in window
regions is much more modest (0.5 dB km$^{-1}$ two-way at 340 GHz for an ice saturated atmosphere at $-30\,^\circ$C, 400 hPa). For example, we are currently developing a short-range 340 GHz radar for in-situ characterisation of snow; similar technology, with a larger antenna and transmit power, could be used to profile cirrus clouds from above.

*Data availability.* The GRaCE G-band data, along with coincident Ka- and W-band data for a range of cases, are described in Courtier et al. (in prep.) and will be publicly accessible from Zenodo in the near future. Until then, the G-band data for the cases shown here can be found



in the supplementary information of this manuscript. The FAAM data used in this paper can be found in the CEDA catalogue (Facility for Airborne Atmospheric Measurements (FAAM), 2024). The HATPRO LWP data is available to download from Cloudnet.

## Appendix A: Derivation of relationships between reflectivity and attenuation by ice

In Sect. 8, we present relationships derived between reflectivity and attenuation by ice. To correct for ice attenuation in this study, we use the relationship in Fig. 7a which was derived between Ka-band reflectivity and attenuation in the G-band, but

we also provide a relationship in Fig. 7b between G-band reflectivity and attenuation in the G-band. Both relationships were derived by performing simulations using the particle models in this study, i.e the four particle mixtures (Large Plate Aggregate mixture, Large Block Aggregate mixture, Large Column Aggregate mixture, and the ICON snow mixture) from the ARTS scattering database (Eriksson et al., 2018) and the three habits (unrimed dendritic aggregates, and rimed dendritic aggregates with ELWP of 0.1 and 0.2 $\mathrm{kgm}^{-2}$) from the database of Mroz and Leinonen (2023). The simulations use PSDs measured in-situ

during a case study on 13/02/2018, which was part of the PICASSO field campaign operating in the region of Chilbolton, UK (i.e. the same geographical region and similar time of year as the case studies considered in this manuscript). A range of PSDs were measured at different altitudes of frontal cloud, at temperatures as low as $-35\,°\mathrm{C}$, using the 2-Dimensional Stereo probe (2D-S; Lawson et al. (2006)) and the High-Volume Precipitation Spectrometer (HVPS-3; Lawson et al. (1993)) instruments on board the FAAM aircraft. The 2D-S is useful for measuring smaller particles in the size range 10 $\mu$m$-1.28$ mm, while

the HVPS-3 is capable of fully measuring large particles up to 19.2 mm. The model particles and measured PSDs were used to calculate dBZ$_{Ka}$, dBZ$_G$, and the specific attenuation by ice in the G-band. Fig. 7 includes green shading which shows the interquartile range of the simulations (representing variations in specific attenuation due to differences in the scattering model and PSDs). The black lines show relationships of the form $f(x) = 10^{ax^2+bx+c}$ that were fit to all the simulated data. The fits allow us to estimate the attenuation by ice given a measured value of dBZ. For $x =$dBZ$_{Ka}$ the coefficients are $a = 3.922\times10^{-6}$,

$b = 8.284 \times 10^{-2}$, $c = -0.8533$, and for $x =$dBZ$_G$ they are $a = 3.618 \times 10^{-4}$, $b = 1.2 \times 10^{-1}$ and $c = 1.492 \times 10^{-2}$.

*Author contributions.* KMC and CW designed the study, performed simulations and data analysis, and wrote the paper. All authors contributed to data interpretation, and contributed to revisions of the text.

*Competing interests.* The authors declare that they have no conflict of interest.

*Acknowledgements.* This work was supported financially by Natural Environment Research Council (NERC; grant numbers NE/V001183/1

and NE/W000946/1). KMC thanks the University of Reading for additional support provided by a Research Endowment Trust Fund grant. The BAe-146 research aircraft is operated by Airtask and Avalon and managed by the Facility for Airborne Atmospheric Measurements (FAAM), which is jointly funded by the Met Office and NERC. The flight was funded by the Met Office. The authors would like to thank the



staff at FAAM and the Met Office who contributed to the data collection. We thank staff at the Chilbolton Observatory for their assistance with maintaining and operating instruments used in this study. Collocated measurements from the 35 GHz Copernicus radar, the HATPRO radiometer, lidar ceilometer and rapid response rain gauge were funded by NERC as part of the National Centre for Atmospheric Science's National Capability science programme. We acknowledge ACTRIS and Finnish Meteorological Institute for providing the HATPRO LWP data.



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
