# Peer review of "Estimating vertical profiles of Ice Water Content and Snowfall Rate from radar measurements in the G-band"

_EGUsphere, 2025_

## Author Comment (AC1)

**Reviewer 1**

This ambitious article investigates the feasibility of retrieving snow ice water content and precipitation rates using vertically-pointed G-band radars. Often, centimeterwavelength radars (S-band, C-band, X-band, Ku-band, etc.) are used for radar retrievals, especially for operational purposes. The authors however find that due to non-Rayleigh scattering effects, ice water content (IWC) and snowfall rate (S) retrievals are expected to vary considerably less at G-band than for these typical centimeter wavelength radars and even other millimeter radars (e.g., Ka- and W-) that are commonly used for ice retrievals – at least for large enough particles. The authors compare the computationally simpler Rayleigh Gans Approximation (RGA) for a number of particles from the ARTS database to the accurate and computationally rigorous Discrete Dipole Approximation (DDA) calculations for the same particles at G-band wavelengths. The authors use these simulations to justify the theoretical power-law scaling of non-Rayleigh scattering that they then use in the rest of the manuscript to derive retrieval equations; this power-law scaling of the parameter 'f' acts as a moment-based integration kernel when calculating IWC and S as well as other moment-based parameters. Overall, the authors show this scaling parameter 'f' leads to IWC and S being directly proportional to G-band Z and Z\*MDV. For suitably sized aggregates, many of the parameters in the IWC and S equations can be treated as constants. The authors use simulations to determine and justify appropriate constant numbers for different types of particles. The authors include another set of numerical experiments where they vary the particle size distribution shapes for various mass-weighted diameters (Dm) and they find that there is only slight variability in results for typical Dm values of snow found near the surface. Finally, the authors utilize G-band data from real snow cases in the United Kingdom where they retrieve IWC and S and then statistically compare results to ground and in-situ measurements.

I found the manuscript to be exceptionally straightforward and easy to read. I thought the experiments were sensible and that the authors took good care of incorporating additional factors such as the impact of attenuation. The biggest limitation of this study really is whether the errors and limitations introduced when utilizing G-band radars are truly worth the benefits provided by the theoretically more accurate retrievals in a more *practical* sense. Overall, I'd like to see a more thorough discussion on the practical aspects and limitations of using vertically-pointed G-band compared to other radar wavelengths. I'm also wondering what the authors' beliefs are regarding how G-band radars should or could be used; should these radars be used only in field campaigns or should they be deployed operationally? Also, I believe the authors should provide rough estimates of expected errors from G-band retrievals compared to similar errors from centimeter and maybe millimeter radars in order for readers to fully appreciate the benefits of utilizing G-band radars. Therefore, I recommend *minor revisions*.

Thank you for your constructive comments on the manuscript!

Major Concerns:

• There is not much discussion regarding the practical limitations of G-band, particularly the role of attenuation and liquid scattering, on snow retrieval errors. Centimeter radars such as the operational C-band radars that are used in Europe would naturally have much less attenuation uncertainty errors in these snow cases. I would imagine that many if not most snow cases in the UK would have wet snow or mixed precipitation. Therefore, it could be the case that G-band radars aren't practical for operational purposes. The authors account for riming in their study but they don't account for liquid coating of snow particles. Do the authors expect their equations to break down considerably for wet snow cases?

Thank you for this comment, and we want to clarify that we do not see G-band as a replacement for scanning cm-wavelength radars of the kind used for operational monitoring of precipitation by national meteorological agencies. G-band is strongly attenuated in the lower atmosphere, and is best suited to vertical profiling. Our focus is accordingly to profile the ice water content and snowfall rates throughout ice clouds and ice-phase precipitation (which does not necessarily involve snowfall at the surface – for example the snowflakes in our second case study evaporate before reaching the ground).

To clarify our intentions here we have inserted the phrase "vertical profiles" into the title of the paper.

• There isn't a discussion regarding what IWC/S errors would be expected from centimeter radars. I think this is necessary because it is hard for readers to judge how much better IWC/S retrievals could be, theoretically, compared to, for example, C-band radars. The authors should consider providing some example calculations of the expected uncertainty ranges (theoretical or using the ARTS database) when using C-band radars compared to the G-band radars. Even a calculation for a single PSD with a fixed Dm and some uncertainty range of Dm would really make it clear to readers how much better G-band radars could be in theory compared to more conventional radar wavelengths.

Thank you for this suggestion – this is useful to include since it is representative of what occurs when Rayleigh scattering is dominant. We have now added simulations at C-band to Figs 1 and 2, and the associated discussion. These show that the ratios IWC/Z and S/(ZxMDV) follow the Rayleigh scattering lines, decreasing continuously with Dm rather than displaying any "flattening" behaviour. In other words, a direct retrieval of IWC or S would not be possible without a-priori assumptions about Dm.

**Minor Concerns:**

• Figure 4: It seems like there is still quite a bit of variability in 'f' for dendritic aggregates and rimed dendritic aggregates (at least a full order of magnitude at the large end). Is this variability significant when considering the propagation of error into IWC and S? I would suggest the authors use kernel density estimates here to better demonstrate how well the fixed power-law relation actually fits the data; the cluster of symbols makes this difficult to tell.

**Thanks for this suggestion -**

We agree that the cluster of symbols was unclear, so we have now changed the way we are plotting the data for unrimed and rimed dendritic aggregates in panels e-g. Instead of plotting individual scatter points for each aggregate, we now show statistical summaries. For each logarithmic bin in  $x = \frac{4\pi R_g}{\lambda}$ , a kernel density estimate is applied to the corresponding values of  $\log_{10} f$ . Quantiles derived from this 1-D KDE are used to form shaded envelopes: the light grey region marks the 10-90% range and the darker grey shows the interquartile range (25-75%). The blue line gives the arithmetic mean of f in each bin, and the red line shows the associated power-law prediction. The natural variability from snowflake to snowflake is smoothed out when averaging over many snowflakes, and the mean fits the power law quite closely. The variability in f is not a significant source of error for retrieving IWC and S, which can be seen by the flattening of the IWC/Z and S/(Z×MDV) lines in Fig 3 to near constant values for Dm>0.5mm.

• Can the authors please describe their vision for how G-band radars should/could be deployed for IWC/S retrievals? The authors state that they expect G-band radars could be used for profiling the bottom 1-km of the atmosphere or perhaps from space-borne platforms. Ka and W band radars are already deployed in spaceborne radars (e.g., GPM-DPR and CloudSat). Should NASA/JAXA consider deploying G-band radars instead? Dm would generally be much smaller at cloud top than at the surface, so maybe G-band radars wouldn't be as desirable for deployment in space?

**Thank you for this comment.**

We show examples (section 8) in which vertical profiles of IWC and S are successfully retrieved. In case 1 the profile is from 0.8-3.8km height; in case 2 it spans the interval 2-7km.

It is true that Dm would be smaller at the cloud top than the surface, and we have discussed in the manuscript that we expect our method to be most useful when Dm>0.5mm which is more likely in precipitating snow, or in low- mid level ice clouds. Unlike passive

infrared measurements, spaceborne radars (like GPM-DPR, CloudSat) sample the full vertical profile, not just the cloud top, so retrievals can be made in this part of the atmosphere. The profiles would still need to be corrected for attenuation along the beam, but measurements from a space-borne platform would experience less attenuation than ground based instruments (because there is much less water vapour in the upper troposphere).

Thus, we envision something similar to CloudSat or EarthCARE but at G-band. We have inserted a sentence in the discussion to clarify this point, in addition to the references to profiling cloud radars in space in the introduction. Following the summary of the case study results we write "This is promising evidence that a G-band spaceborne radar sampling vertical profiles of the kind that are currently being sampled by EarthCARE would provide valuable observations of the vertical profiles of ice in clouds and snow falling close to the surface."

Suggestions/Typos/etc.:

Lines 6-7: The sentence should read: "This presents the opportunity for straightforward *and* accurate retrievals of ice microphysics."

"Straightforward" and "accurate" retrievals are both positive attributes.

Fixed this.

Line 59: It is more appropriate to say that Z and IWC are proportional to PSD moments rather than they are the moments themselves.

Fixed this.

Line 117: There should be a space between 2 and mm.

Fixed this.

Line 177: Inconsistent usage of figure reference. This should probably say "Figs 1 and 2" similar to line 220 rather than spelling out "figure."

Fixed this.

---

## Author Comment (AC2)

**Reviewer 2**

This manuscript presents theory and simulations for deriving Ice Water Content (IWC) and Snowfall Rate S directly from stand-alone G-band radar reflectivity and Doppler velocity measurements. The derived retrieval is applied to two case studies with measurements from the GRaCE 200GHz Doppler radar at Chilbolton. Retrieved IWC and S are compared to airborne in-situ IWC and surface gauge measurements, respectively.

The presented retrieval is convincingly introduced by concise simulation and theory. Limitations are well discussed when applied to measurements. The manuscript is timely as it, for the first time, makes use of G-band radar data to derive cold microphysics parameters. I recommend publication with *minor revisions* after the following comments have been addressed by the authors.

**Thanks a lot for your helpful feedback on the manuscript!**

**General comments**

• The manuscript contains 9 Sections. In order to make the manuscript structure more compact and highlight connections between the Sections, I recommend moving Sections 5-7 as subsections to Section 4, or combining them in a new Section 5, e.g. labeled: Sensitivity to retrieval parameters. For similar reasons, I propose to move the description and discussion of the attenuation correction in Sec 8 (II 338-366) to a stand-alone subsection.

We have now added a stand-alone subsection (8.1) to discuss the attenuation corrections considered when looking at the data collected during the two case studies. We appreciate your suggestion of combining Sections 5-7 to make the structure more compact and have given it careful consideration. However, we were unable to come up with a meaningful way to do this as the content in these sections does not all fall under the category of sensitivity analysis. For example, sections 5 and 6 connect the simulation data to the theory. As a practical part of that we see what happens for different scattering models, but the purpose is not purely to vary the scattering model and see the differences.

 The authors compare retrieved IWC and S to rain-gauge and in-situ data, respectively. Here, it would be nice if the G-band performance could be further highlighted compared to state-of-the-art empirical relations obtained from "standard" radars operating at Ka- or W-band, to give the reader an idea on advantages compared to other cloud radars.

Thank you for this idea to further emphasise the value of our new technique. We did begin to explore this idea; however we have ultimately chosen not to include a comparison with empirical relationships at low frequencies. Our reasoning is as follows:

The best way to understand the advantage of G-band is expressed in figures 1 and 2, which show that unless Dm is known a-priori, there are large uncertainties in estimating IWC and S from radar data at lower frequencies. Empirical relationships between these variables and radar parameters at "standard" frequencies like Ka or W band therefore have to rely (either explicitly or implicitly) on statistical correlations between Dm and Z in order to make a retrieval. These relationships can vary between different geographical regions, different cloud types, even between different regions of a single heterogenous cloud system. At G-band things are much less uncertain, because the quantity that you measure and the microphysical property you want to know are essentially proportional to the same moment of the size distribution.

We did begin to compare some empirical IWC-Z and S-Z relationships to our retrievals in the case studies. We found some similar structures in the data and some qualitative and quantitative differences (some of them large). The challenge then is to interpret what those differences mean. It could reflect the weakness in what these empirical relationships assume about the size distribution parameters, as discussed above, highlighting the benefit of G-band which is insensitive to those issues. But it could also reflect differences in what is assumed about the characteristics of the particles: e.g. a different mass-size relationship inconsistent with our choices.

The other issue is that our verification data is imperfectly co-located with our radar samples, and this means that there will always be deviations from any retrieval, even if it were perfect. It is good enough to make an assessment that our retrievals are realistic, but it makes a meaningful comparison of two competing retrievals against the gauge / aircraft data difficult.

So, although we agree that this is an interesting practical question, the interpretation of the comparison is more subtle than it first appears, and opens up a number of non-trivial questions which are beyond the scope of this paper, and which we feel would distract from our key findings.

**Minor comments:**

- Fig 1: plotted variable name should be added to y-axis label, also L105, 108
  Fixed this.
- Figs 9c, d; 10; 13: missing label on colorbars
  Fixed this.
- Fig 12 caption: description of colors missing
  Fixed this.

**Technical Comments:**

• L1-4: sentence is very long. I suggest to split into two: [...] proportional to their mass (m). Hence, measurements [...]

Fixed this.

• L124: snowfall rate S (italics)

Fixed this.